# RNF185 regulates proteostasis in Ebolavirus infection by crosstalk between the calnexin cycle, ERAD, and reticulophagy

Jing Zhang[1], Bin Wang[1,2], Xiaoxiao Gao[3], Cheng Peng[3], Chao Shan [3,4], Silas F. Johnson [5], Richard C. Schwartz [5] & Yong-Hui Zheng [1,5] ✉

Virus infection affects cellular proteostasis and provides an opportunity to study this cellular process under perturbation. The proteostasis network in the endoplasmic reticulum (ER) is composed of the calnexin cycle, and the two protein degradation pathways ER-associated protein degradation (ERAD) and ER-to-lysosome-associated degradation (ERLAD/ER-phagy/reticulophagy). Here we show that calnexin and calreticulin trigger *Zaire* Ebolavirus (EBOV) glycoprotein $GP_{1,2}$ misfolding. Misfolded $EBOV-GP_{1,2}$ is targeted by ERAD machinery, but this results in lysosomal instead of proteasomal degradation. Moreover, the ER Ub ligase RNF185, usually associated with ERAD, poly-ubiquitinates $EBOV-GP_{1,2}$ on lysine 673 via ubiquitin K27-linkage. Poly-ubiquinated $GP_{1,2}$ is subsequently recruited into autophagosomes by the soluble autophagy receptor sequestosome 1 (SQSTM1/p62), in an ATG3- and ATG5-dependent manner. We conclude that EBOV hijacks all three proteostasis mechanisms in the ER to downregulate $GP_{1,2}$ via polyubiquitination and show that this increases viral fitness. This study identifies linkages among proteostasis network components previously thought to function independently.

The endoplasmic reticulum (ER) serves as the entry site of the secretory pathway, which synthesizes ~30% of all proteins in eukaryotic cells[1]. Thus, the ER must ensure the quality of native polypeptides by promoting their folding and degrading aberrant gene products via a proteostasis-network that consists of protein folding cycles and two protein degradation pathways. In the ER, protein disulfide isomerase family A member 3 (PDIA3/ERp57) interacts with calnexin (CANX) and calreticulin (CALR) to comprise the calnexin cycle, which promotes oxidative folding of *N*-glycosylated glycoproteins by mediating disulfide bond formation[2]. Notably, protein folding in the ER is inefficient and error-prone, resulting in accumulation of misfolded proteins that disrupt ER proteostasis network[3]. Misfolded proteins can be cleared by proteasomes via ER-associated degradation (ERAD), or by lysosomes via ER-to-lysosome-associated degradation (ERLAD), known as ER-phagy or reticulophagy[4]. ERAD dislocates aberrant proteins from the ER lumen to the cytosolic ubiquitin proteasome system for degradation. Reticulophagy encapsulates an ER subdomain containing misfolded proteins for lysosomal clearance via reticulophagy receptors expressed on the ER membrane. Compared to ERAD, the mechanism of reticulophagy is still poorly understood. For example, mammalian cells express over 600 ubiquitin (Ub) E3 ligases to specifically target different substrates for polyubiquitination[5], and at least 25 of them are embedded on the ER membrane[6]. A dozen of these ER Ub ligases, such as tripartite motif containing 13 (TRIM13)[7], Really Interesting New Gene

[1]Harbin Veterinary Research Institute, CAAS-Michigan State University Joint Laboratory of Innate Immunity, State Key Laboratory of Veterinary Biotechnology, Chinese Academy of Agricultural Sciences, Harbin, China. [2]MSD (Ningbo) Animal Health Technology Co., Ltd., Ningbo, China. [3]Center for Biosafety Mega-Science, Wuhan Institute of Virology, Chinese Academy of Sciences, Wuhan, China. [4]State Key Laboratory of Virology, Wuhan Institute of Virology, Chinese Academy of Sciences, Wuhan, China. [5]Department of Microbiology and Molecular Genetics, Michigan State University, East Lansing, MI, USA. ✉e-mail: zhengyo@msu.edu

(RING) finger protein 26 (RNF26)[6], and RNF185[8], have been identified to polyubiquitinate misfolded proteins during ERAD[9]. However, it is still unclear whether these E3 Ub ligases polyubiquitinate any aberrant proteins during reticulophagy. In addition, it is unclear whether the components in the calnexin cycle, ERAD, and reticulophagy are pathway-specific or they are shared in this network.

Filoviruses are enveloped viruses with a negative-sense, single-stranded RNA genome comprising seven genes[10]. Two genera from this virus family, *Ebolavirus* and *Marburgvirus*, are highly pathogenic, and cause severe hemorrhagic fever in humans and nonhuman primates with up to a 90% mortality rate[11]. Filovirus entry is mediated by the virion-associated structural glycoprotein $GP_1$/$GP_2$ trimer, which is proteolytically processed in the Golgi apparatus from $GP_{1,2}$ as a precursor. $GP_{1,2}$ is expressed in the ER from the fourth gene of the viral genome, *GP*[12]. Unlike $GP_{1,2}$ from Marburg virus (MARV), high levels of $GP_{1,2}$ expression from *Zaire* ebolavirus (EBOV) triggers cell rounding, detachment, and downregulation of many surface molecules and these cellular effects may contribute to its high pathogenicity[13–18]. The determinants for these cellular effects are the mucin-like domain (MLD) in $GP_1$ and the transmembrane domain in $GP_2$ of EBOV[16,18,19]. To balance these cellular effects against productive viral infection and spread, Ebolaviruses reduce their $GP_{1,2}$ expression up to 70% at the mRNA level by an RNA editing mechanism, with the vast majority of *GP* mRNAs being used to produce nonstructural soluble GP (sGP) and a small amount of small soluble GP (ssG*P*)[20,21]. The redundant sGP proteins promote immune evasion by competing for neutralizing $GP_{1,2}$-specific antibodies[22]. The sGP proteins also increase virus uptake into late endosomes by activating the mitogen-activated protein kinase (MAPK) signaling pathway[23]. Thus, a low-level of $GP_{1,2}$ expression and a high-level of sGP expression are beneficial to *Ebolavirus* by promoting viral fitness and immune evasion. We recently reported that $GP_{1,2}$ expression is further downregulated by PDIs from the ER[24], which serve as another important pathway to lower the $GP_{1,2}$ expression.

We now report that the CANX-CALR cycle is involved in EBOV-$GP_{1,2}$ misfolding via PDIA3, and that misfolded $GP_{1,2}$ proteins are targeted by ERAD machinery (class I mannosidases, valosin-containing protein (VCP)/p97) and degraded in lysosomes. $GP_{1,2}$ degradation is dependent on autophagy-related (ATG) genes *ATG3* and *ATG5* that are required for microtubule-associated protein 1 light chain 3 (MAP1LC3/ LC3) lipidation. In addition, $GP_{1,2}$ degradation is also dependent on the soluble autophagy receptor sequestosome 1 (SQSTM1/p62) that recruits polyubiquitinated cargoes. Importantly, we identified RNF185, as the key Ub ligase that polyubiquitinates EBOV-$GP_{1,2}$ on lysine 673 (K673) in its cytoplasmic tail via K27-linked ubiquitination. Thus, EBOV hijacks all three proteostasis network mechanisms in the ER to downregulate $GP_{1,2}$ via polyubiquitination, uncovering unexpected linkages in the ER proteostasis-network that were traditionally considered to function independently.

## Results

### CANX and CALR downregulate steady state ebolavirus $GP_{1,2}$ expression

We identified PDIA3 as an EBOV-$GP_{1,2}$ interacting protein in the ER by liquid chromatography tandem mass spectrometry (LC-MS/MS), resulting in discovery of EBOV-$GP_{1,2}$ degradation via reticulophagy[24]. In the same study, we also identified three ER chaperones, CALR, CANX, and Heat Shock Protein Family A (Hsp70) Member 5 (HSPA5), that interacted with EBOV-$GP_{1,2}$. To understand how these chaperones affect EBOV-$GP_{1,2}$ expression, FLAG-tagged $GP_{1,2}$ was expressed with HA-tagged PDIA3, HSPA5, CALR, or CANX in HEK293T cells, and their steady state expression was analyzed by western blotting (WB). $GP_{1,2}$ expression was strongly reduced by PDIA3, CALR, and CANX, whereas HSPA5 did not have any effect (Fig. 1A). We previously reported that PDIA3 also decreases expression of $GP_{1,2}$ with a MLD-deletion

(GPΔMLD), but not sGP or ssGP[24]. When CALR and CANX were expressed with these different GP forms, CANX decreased GPΔMLD expression, whereas CALR did not (Fig.1B, lane 3, 4, 11, 12), and neither decreased sGP nor ssGP expression (Fig. 1B, lanes 5–8, 13–16). When GP interactions with CALR and CANX were tested by co-immunoprecipitation (IP), $GP_{1,2}$, GPΔMLD, sGP, and ssGP all interacted with both CALR and CANX (Fig. 1C). These results demonstrate that CALR and CANX selectively downregulate $GP_{1,2}$ expression, although they interact with all different GPs.

To validate these results, we tested the endogenous CALR and CANX activity on $GP_{1,2}$ expression. When $GP_{1,2}$ was expressed in HEK293T wild-type (WT), and *PDIA3*-, *CALR*-, or *CANX*-knockout (KO) cells, $GP_{1,2}$ levels were much higher in all three KO cell lines than in the WT cell line, confirming the CALR and CANX inhibitory activity (Fig. 1D). CALR and CANX activity were further confirmed by their dose-dependent inhibition of $GP_{1,2}$ expression in WT and their KO cell lines (Fig. 1E). Next, we tested CALR and CANX activity against $GP_{1,2}$ proteins from the other ebolaviruses, including *Sudan* ebolavirus (SBOV), *Bundibugyo* ebolavirus (BDBV), Taï Forest ebolavirus (TAFV), and Reston ebolavirus (RESTV). Expression of these different $GP_{1,2}$ proteins was strongly reduced by both ectopic (Fig. 1F) and endogenous (Fig. 1G) CALR and CANX. Notably, when $GP_{1,2}$ from another filovirus, MARV, was tested, its expression was strongly increased by both ectopic and endogenous CALR and CANX (Fig. 1H). When CALR and CANX activity were further tested in several other cell lines, including A549, HeLa, Hep G2, mouse primary macrophages (MΦ), and human monocytic THP1 cell-derived macrophages (THP1-MΦ), they reduced $GP_{1,2}$ expression to a similar level as in HEK293T cells (Fig. 1I). Collectively, these results demonstrate that CALR and CANX downregulate ebolavirus $GP_{1,2}$ steady state expression in a cell type-independent manner.

### CANX and CALR decrease EBOV entry

To understand the functional impacts of CANX and CALR activity on EBOV-$GP_{1,2}$ expression, we used viral assays to test how EBOV entry is affected by these two ER chaperones. Initially, we used HIV-1 pseudo-viruses that expressed EBOV-$GP_{1,2}$ to measure EBOV entry, which is a standard assay for studying $GP_{1,2}$ activity[24–26]. Envelope glycoprotein (Env)-deficient HIV-1 luciferase reporter viruses were produced in the presence of EBOV-$GP_{1,2}$ from HEK293T WT, and PDIA3-, CALR-, or CANX-overexpressing, and *PDIA3*-, *CALR*-, or *CANX*-KO cells. An equal amount of these pseudoviruses was collected to infect HEK293T cells, and viral entry was determined by measuring intracellular luciferase activity. Like PDIA3, ectopic CANX and CALR expression strongly decreased, whereas their KOs strongly increased EBOV entry (Fig. 2A). Next, we tested whether their inhibitory activity could be recapitulated with EBOV replication- and transcription-competent virus-like particles (trVLPs), which model the entire viral replication cycle[27]. Like PDIA3, both CALR and CANX consistently suppressed EBOV replication more than 10-fold in three consecutive passages (p0, p1, p2) in HEK293T cells (Fig. 2B). Importantly, PDIA3, CALR, and CANX also inhibited replication of the authentic EBOV Mayinga strain in Huh7 cells in a dose-dependent manner (Fig. 2C). Furthermore, we tested how they affect $GP_{1,2}$ levels in virions. EBOV virus-like particles (VLPs) were produced from CALR- or CANX-overexpressing, and *CALR*- or *CANX*-KO cells after expressing EBOV-$GP_{1,2}$ with its matrix protein VP40. Both ectopic and endogenous CALR and CANX reduced $GP_{1,2}$ levels in these VLPs, although the endogenous protein activity was slightly weaker than the ectopic protein activity (Fig. 2D, lower panels). Finally, we tested how CALR and CANX affect expression of glycoproteins from 6 other enveloped viruses, including influenza A virus (IAV)-hemagglutinin (HA), vesicular stomatitis virus (VSV)-glycoprotein (G), HIV-1 Env, Middle East respiratory syndrome coronavirus (MERS-CoV) spike protein (S), severe acute respiratory syndrome coronavirus (SARS-CoV) S, and SARS-CoV-2 S proteins. Neither ectopic

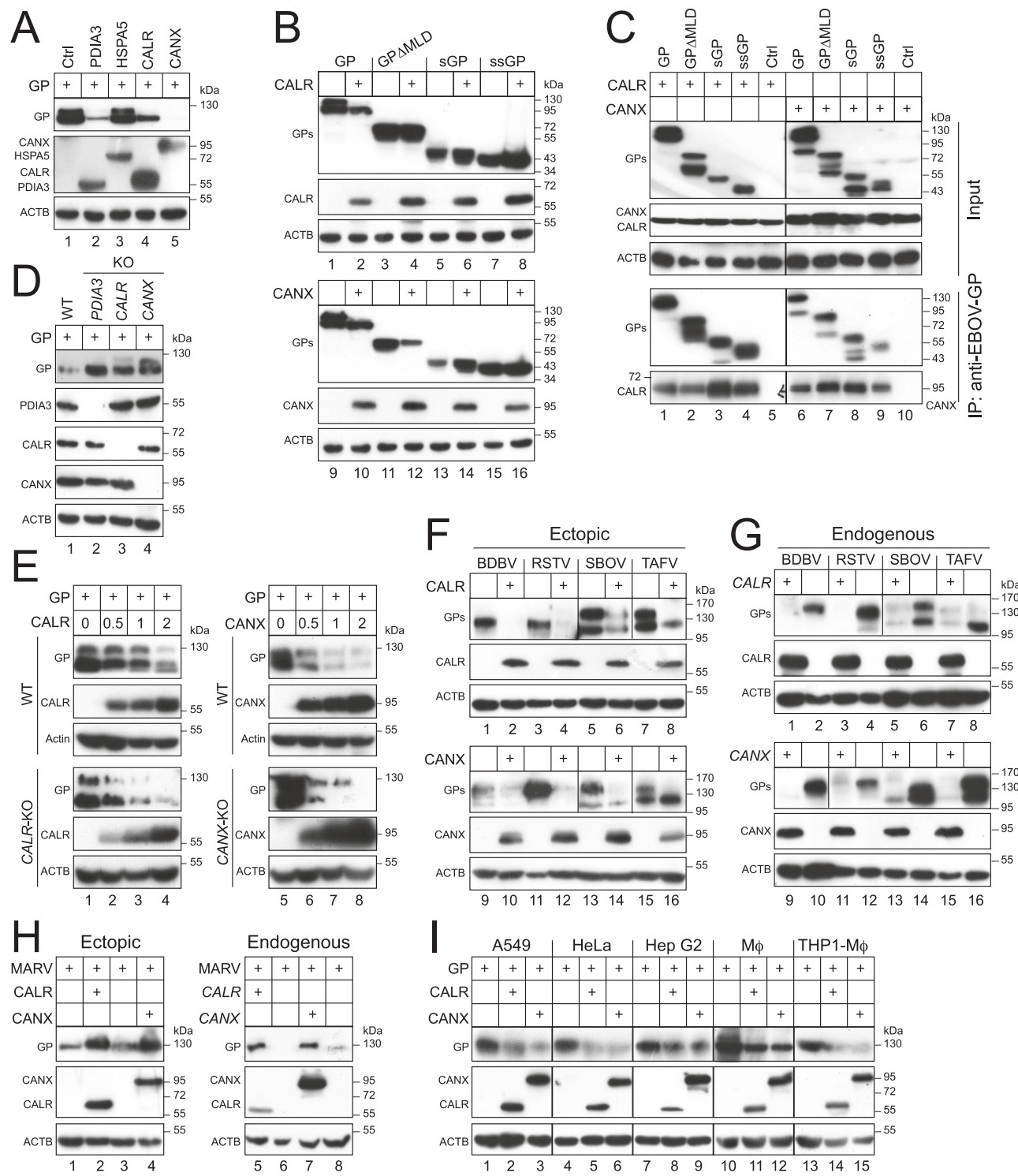

nor endogenous CALR and CANX had any noticeable effect (Fig. 2E and Fig. 2F, respectively), except that ectopic CALR increased VSV-G and HIV-1 Env expression (Fig. 2E, lane 5 to 8). These results demonstrate that CALR and CANX decrease EBOV entry by reducing its $GP_{1,2}$ levels in virions, which is consistent with their downregulation of $GP_{1,2}$ in viral producer cells.

**Interrelationship between CANX, CALR, and PDIA3 in EBOV-$GP_{1,2}$ downregulation**

CANX and CALR interact with PDIA3 and promote glycoprotein folding in the ER[28,29]. To understand their interrelationship in EBOV-$GP_{1,2}$ downregulation, we created single *PDIA3*-, *CALR*-, or *CANX*-KO, and

dual *PDIA3/CANX*- or *PDIA3/CALR*-KO cell lines from HEK293T cells (Fig. 3A). Next, EBOV-$GP_{1,2}$ was expressed with HSPA5, PDIA3, CALR, and CANX in these cells, and levels of $GP_{1,2}$ expression were compared. HSPA5 barely affected $GP_{1,2}$ expression in any of these cells (Fig. 3B, lanes 3, 8, 13, 18, 23). In contrast, PDIA3 (Fig. 3B, lanes 2, 7, 12, 17, 22) and CANX (Fig. 3B, lanes 5, 10, 15, 20, 25) strongly decreased $GP_{1,2}$ expression in all these cells. However, CALR decreased $GP_{1,2}$ expression much less effectively in the single *PDIA3*- or *CANX*-KO, and dual *PDIA3/CANX*- or *PDIA3/CALR*-KO cells, although an effective decrease was detected in the single *CALR*-KO cells (Fig. 3B, lanes 4, 9, 14, 19, 24). These results demonstrate that CALR is dependent on CANX and PDIA3, while CANX is independent of CALR and PDIA3 in

**Fig. 1 | CANX and CALR decrease ebolavirus GP$_{1,2}$ expression at steady state.**
**A** EBOV-GP$_{1,2}$ (GP) was expressed with indicated ER proteins in HEK293T cells. Protein expression was detected by western blotting (WB). GP was detected by anti-EBOV-GP; PDIA3 and CALR were detected by anti-Myc; HSPA5 was detected by anti-FLAG; CANX was detected by anti-HA. Ctrl, control vector; ACTB, β-actin. **B** Indicated EBOV glycoproteins were expressed with CALR or CANX in HEK293T cells. Protein expression was detected by WB. EBOV glycoproteins were detected by anti-HiBiT; CALR was detected by anti-Myc; CANX was detected by anti-HA. **C** Indicated EBOV glycoproteins were expressed with CANX or CALR in HEK293T cells, and co-immunoprecipitated (IP) with anti-EBOV-GP from cell lysate (Input). The GP interactions with CANX and CALR were determined by WB. GPs were detected by anti-EBOV-GP; CALR was detected by anti-Myc; CANX was detected by anti-HA. **D** EBOV-GP$_{1,2}$ was expressed in HEK293T wild-type (WT) and *PDIA3-*, *CALR-*, or *CANX*-knockout (KO) cells. Protein expression was detected by WB. GP and endogenous PDIA3, CALR, and CANX were detected by their specific antibodies. **E** EBOV-GP$_{1,2}$ was expressed with increased amounts of CALR or CANX in HEK293T WT and *CALR-* or *CANX*-KO cells. Protein expression was detected by WB. GP was detected by anti-EBOV-GP; CALR and CANX were detected by anti-Myc or anti-HA. **F** GP$_{1,2}$ proteins from indicated ebolaviruses were expressed with CALR or CANX in HEK293T cells. Protein expression was detected by WB. GPs were detected by anti-EBOV-GP; CALR and CANX were determined by anti-Myc or anti-HA. **G** GP$_{1,2}$ proteins from indicated ebolaviruses were expressed in HEK293T WT and *CALR-* or *CANX*-KO cells. Protein expression was detected by WB. GPs were detected by anti-EBOV-GP; CALR and CANX were detected by their specific antibodies. **H** MARV-GP$_{1,2}$ was expressed with CANX or CALR in HEK293T cells. Alternatively, MARV-GP$_{1,2}$ was expressed in HEK293T WT and *CALR-* or *CANX*-KO cells. Protein expression was detected by WB. MARV-GP was detected by anti-HiBiT; ectopic CALR and CANX were detected by anti-Myc or anti-HA; endogenous CALR and CANX were detected by their specific antibodies. **I** EBOV-GP$_{1,2}$ was expressed with CANX or CALR in A549 cells, HeLa cells, Hep G2 cells, primary mouse macrophages (Mφ), and THP1-derived macrophages (THP1-Mφ). Protein expression was detected by WB. GP was detected by anti-EBOV-GP; CALR and CANX were detected by anti-Myc or anti-HA.

downregulating GP$_{1,2}$ expression. To further understand the role of PDIs in this downregulation, cells were treated with Tizoxanide (TIZ), which specifically inhibits the PDIA3 disulfide reductase activity[30]. Notably, GP$_{1,2}$ downregulation by PDIA3, CALR, and CANX was inhibited by TIZ in a dose-dependent manner for all three proteins (Fig. 3C). Thus, although the CANX activity was independent of PDIA3, it still requires a disulfide reductase activity to downregulate GP$_{1,2}$.

## CANX and CALR target EBOV-GP$_{1,2}$ to the autolysosomes for degradation via ERAD machinery

To understand how CANX and CALR downregulate EBOV-GP$_{1,2}$ expression, we blocked protein degradation pathways with their respective inhibitors. First, we blocked ERAD with three inhibitors, including kifunesine (Kif), eeyarestatin I (EerI), and lactacystin (Lac). Kif inhibits the class I α-mannosidases that initiate ERAD; EerI inhibits VCP/p97, a member of the AAA + ATPase family that is required for retro-transportation of ERAD client proteins from the ER to the cytoplasm; and Lac inhibits proteasomes. GP$_{1,2}$ downregulation by CALR and CANX was blocked by Kif and EerI but not Lac (Fig. 4A), indicating that class I α-mannosidases and VCP/p97, but not proteasomes, are required. Second, we blocked the autolysosome activity with bafilomycin A1 (BafA1), concanamycin A (ConA), and NH$_4$Cl, and the proteasome activity again with MG132 and Lac. CALR and CANX activities were both blocked by autolysosome inhibitors, but not proteasome inhibitors (Fig. 4B). Third, we blocked autophagosome synthesis by 3-methyladenine (3-MA), LY 294002 (LY), and wortmannin (Wort). CALR and CANX activities were both blocked by these autophagy inhibitors (Fig. 4C).

In mammalian cells, there are three primary types of autophagy: macroautophagy/autophagy, chaperone-mediated autophagy (CMA), and microautophagy[31]. Macroautophagy/autophagy is mediated by autophagosomes, whereas microautophagy is not, and CMA requires LAMP2A (lysosomal-associated membrane protein 2A) as a receptor. When *LAMP2A* was silenced by small interfering RNAs (siRNAs), CALR and CANX activities were not affected (Fig. 4D). Formation of autophagosomes is dependent on a number of ATG proteins, which induce lipidation of soluble MAP1LC3/LC3-I to become LC3-II on phagophore membranes[32]. When LC3-II synthesis was blocked in HEK293T cells by knocking out either *ATG3* or *ATG5*, CALR and CANX activities were both blocked (Fig. 4E). We further knocked out a soluble autophagy receptor SQSTM1/p62 in HEK293T, HeLa, and A549 cells, and another autophagy receptor, histone deacetylase 6 (HDAC6), in A549 cells. CANX and CALR activities were both blocked in all three *SQSTM1*-KO cell lines (Fig. 4F), and none of them were blocked in the *HDAC6*-KO cell line (Fig. 4G). Thus, EBOV-GP$_{1,2}$ is recruited to autophagosomes in a SQSTM1-dependent, but HDAC6-independent manner, in the presence of CANX and CALR.

To confirm that EBOV-GP$_{1,2}$ is degraded in autolysosomes, we measured the GP$_{1,2}$ turnover rate in the presence and absence of CANX and CALR after blocking cellular translation by cycloheximide (CHX). GP$_{1,2}$ had a half-life of ~10 h, which was decreased to less than 2 h by CANX and CALR (Fig. 4H). This decrease was restored by treatment with BafA1 (Fig. 4H). We then used confocal microscopy to determine GP$_{1,2}$ localization in autolysosomes using LAMP1 as a marker. GP$_{1,2}$ and LAMP1 co-localization was clearly detected in the presence of CANX and CALR, which was further enhanced by treatments with BafA1 or NH$_4$Cl (Fig. 4I). Collectively, these results demonstrate that CANX and CALR target EBOV-GP$_{1,2}$ to the autolysosomes for degradation. Importantly, class I α-mannosidases and VCP/p97 that play an important role in ERAD are required for the entry of clients into reticulophagy.

## RNF26 decreases EBOV-GP$_{1,2}$ expression in an E3 ubiquitin ligase activity-independent manner

Our results showed that SQSTM1/p62 is required for EBOV-GP$_{1,2}$ degradation via autophagy. To be an active autophagy receptor, SQSTM1/p62 needs to be polyubiquitinated[33]. In fact, SQSTM1/p62 is polyubiquitinated by RNF26, which also tethers endosomes to the ER[34]. In addition, RNF26 recruits the Ub conjugating enzyme E2 J1 (UBE2J1) that plays an important role in ERAD[35,36]. Thus, we investigated the role of RNF26 in EBOV-GP$_{1,2}$ polyubiquitination and degradation.

Initially, we investigated whether RNF26 targets EBOV-GP$_{1,2}$. To confirm the ER localization of RNF26, RNF26 with a C-terminal mCherry tag was expressed in HeLa cells with the ER marker CALR or the Golgi marker TGOLN2 (Trans-Golgi network integral membrane protein 2), both of which have a C-terminal GFP tag. RNF26 co-localized with CALR, but not TGOLN2, when detected by confocal microscopy (Fig. 5A). To detect the RNF26 and EBOV-GP$_{1,2}$ interaction, EBOV-GP$_{1,2}$ with a C-terminal GFP tag was expressed with RNF26-mCherry in HeLa cells, and their co-localization was determined by confocal microscopy. EBOV-GP$_{1,2}$ alone was detected on the plasma membrane, but was found in the cytoplasm in the presence of RNF26 (Fig. 5B). Importantly, strong EBOV-GP$_{1,2}$ and RNF26 co-localization was clearly detected, indicating that their interaction occurred in the ER. In addition, ectopic RNF26 strongly decreased the expression of GP$_{1,2}$ from EBOV, RSTV, SBOV, and TAFV in HEK293T cells (Fig. 5C). Thus, RNF26 targets ebolavirus GP$_{1,2}$ and decreases its expression.

Next, we investigated how RNF26 targets EBOV-GP$_{1,2}$. RNF26 has 433 amino acids (aa) that comprise five transmembrane (TM) domains in the N-terminal half region and a RING-finger in the C-terminal region (Fig. 5D). We created five RNF26 deletion mutants to express its different regions that cover 76–433, 241–433, 1–385, 76–385, and 241–385 aa, and four point-mutation mutants C395S, C399S, C401S, and 3C/3S

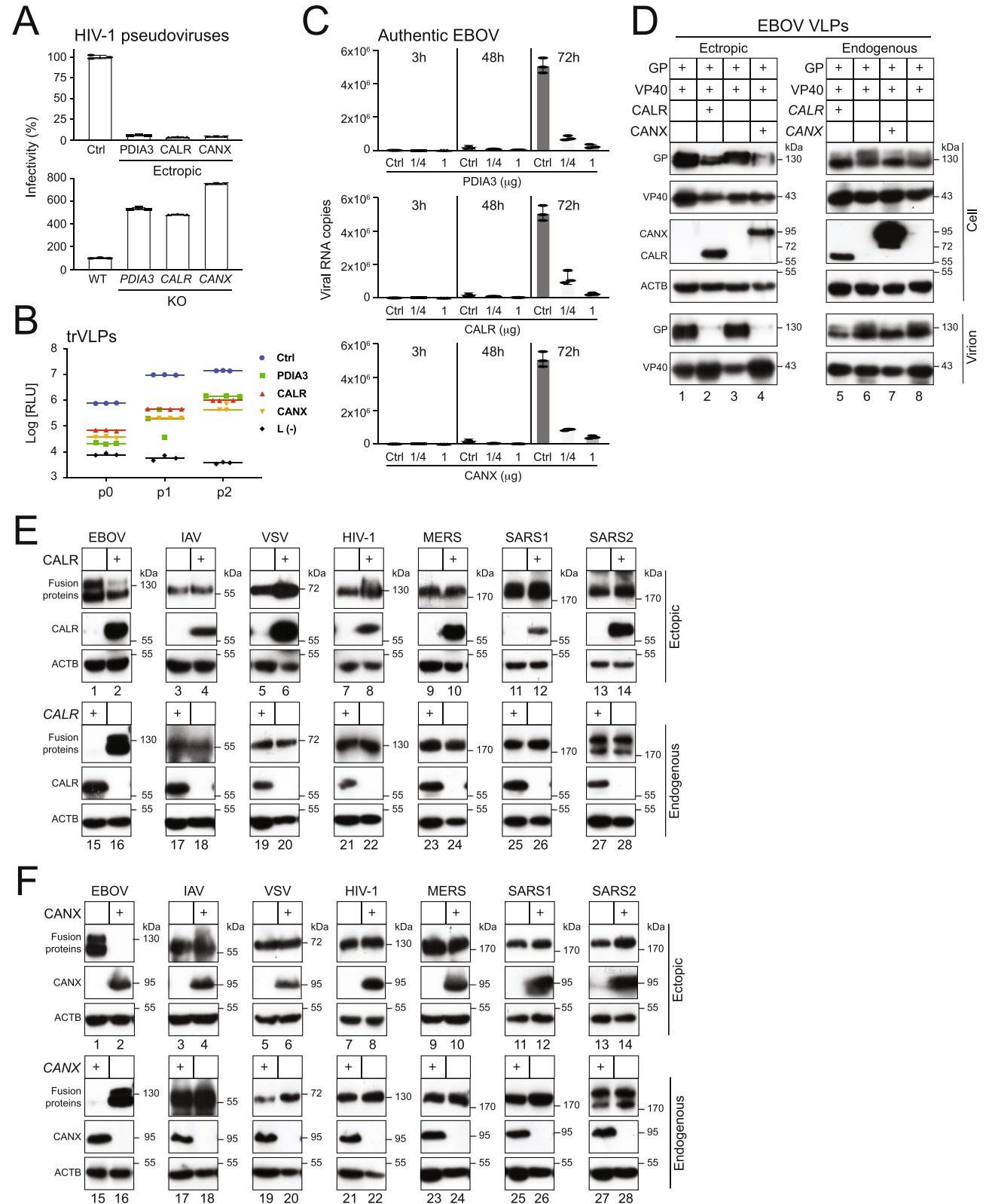

that replace the critical Cys residues in the RING-finger with Ser (Fig. 5D). When RNF26 WT and these mutants were expressed with EBOV-GP$_{1,2}$ in HEK293T cells, all these RNF26 proteins were expressed at a similar level (Fig. 5E). However, unlike the others, mutants 241–433 and 241–385 did not decrease GP$_{1,2}$ expression (Fig. 5E, lanes 4, 7). As mutants C395S, C399S, C401S, 3C/3S, and 1–385 were still active, we conclude that the RING-finger is not required for this RNF26 activity. In addition, because mutant 76–385 was still active, whereas mutant 241–385 was not, we conclude that the 76–240 region that contains TM3, TM4, and TM5 domains is required for the RNF26 activity.

Next, we investigated how RNF26 interacts with EBOV-GP$_{1,2}$ by co-IP. EBOV-GP$_{1,2}$ was expressed with RNF26 WT and its mutants 76–433, 241–433, 1–385, 76–385, and 241–385, and RNF26 proteins were pulled down to detect EBOV-GP$_{1,2}$. RNF26 WT and its mutants 76–433, 1–385, and 76–385 pulled down EBOV-GP$_{1,2}$, whereas mutants 241–433 and 241–385 did not (Fig. 5F). These results confirm the RNF26 interaction

**Fig. 2 | CANX and CALR reduce EBOV entry. A** HIV-1 firefly luciferase reporter viruses pseudotyped with EBOV-GP$_{1,2}$ were produced from HEK293T WT, PDIA3, CALR or CANX-overexpressing, and *PDIA3*-, *CALR*- or *CANX*-KO cells. After infecting HEK293T cells with an equal number of these different viruses, viral entry was determined by measuring intracellular luciferase activities. Viral entry is shown as relative values, with the entry of viruses produced from HEK293T WT cells in the presence of a control vector set to 100%. **B** EBOV replication and transcription-competent virus-like particles (trVLPs) were produced (p0) and passaged two times (p1, p2) in HEK293T cells in the presence or absence of PDIA3, CALR, or CANX. EBOV replication was determined by measuring intracellular *Renilla* luciferase activity. Viral replication was also measured in the absence of EBOV-L [L(-)], which served as a negative control. Results from three independent experiments are presented. **C** Huh7 cells were transfected with increasing amounts of vectors expressing PDIA3, CALR, or CANX, and infected with EBOV Mayinga strain at 0.01 multiplicity of infection (MOI). Viral RNAs were extracted from the supernatants at indicated times and quantified by real-time PCR. **D** EBOV virus-like particles (VLPs)

were produced from HEK293T WT, CALR or CANX-overexpressing, and *CALR*- or *CANX*-KO cells after expression of EBOV-GP$_{1,2}$ with EBOV-VP40. VLPs were purified by ultra-centrifugation and protein expression in cell lysate and virions was analyzed by WB. GP, VP40, and endogenous CALR and CANX were detected by their specific antibodies; ectopic CALR and CANX were detected by anti-Myc or anti-HA. **E** Viral fusion proteins from indicated viruses were expressed with CALR in HEK293T cells (lanes 1–14). Alternatively, they were also expressed in HEK293T WT or *CALR*-KO cells (lanes 15-28). Protein expression was detected by WB. EBOV-GP$_{1,2}$, MERS-S, SRAS1-S, and SARS2-S were detected by anti-FLAG; IAV (H5N5) HA, VSV-G, HIV-1 Env, and CALR were detected by their specific antibodies. **F** Viral fusion proteins from indicated viruses were expressed with CANX in HEK293T cells (lanes 1-14). Alternatively, they were also expressed in HEK293T WT or *CANX*-KO cells (lanes 15-28). Protein expression was detected by WB as in **D**, except that CANX was detected by its specific antibody. Error bars in **A**, **B**, and **C** represent the standard error of measurements (SEMs) calculated from three independent experiments.

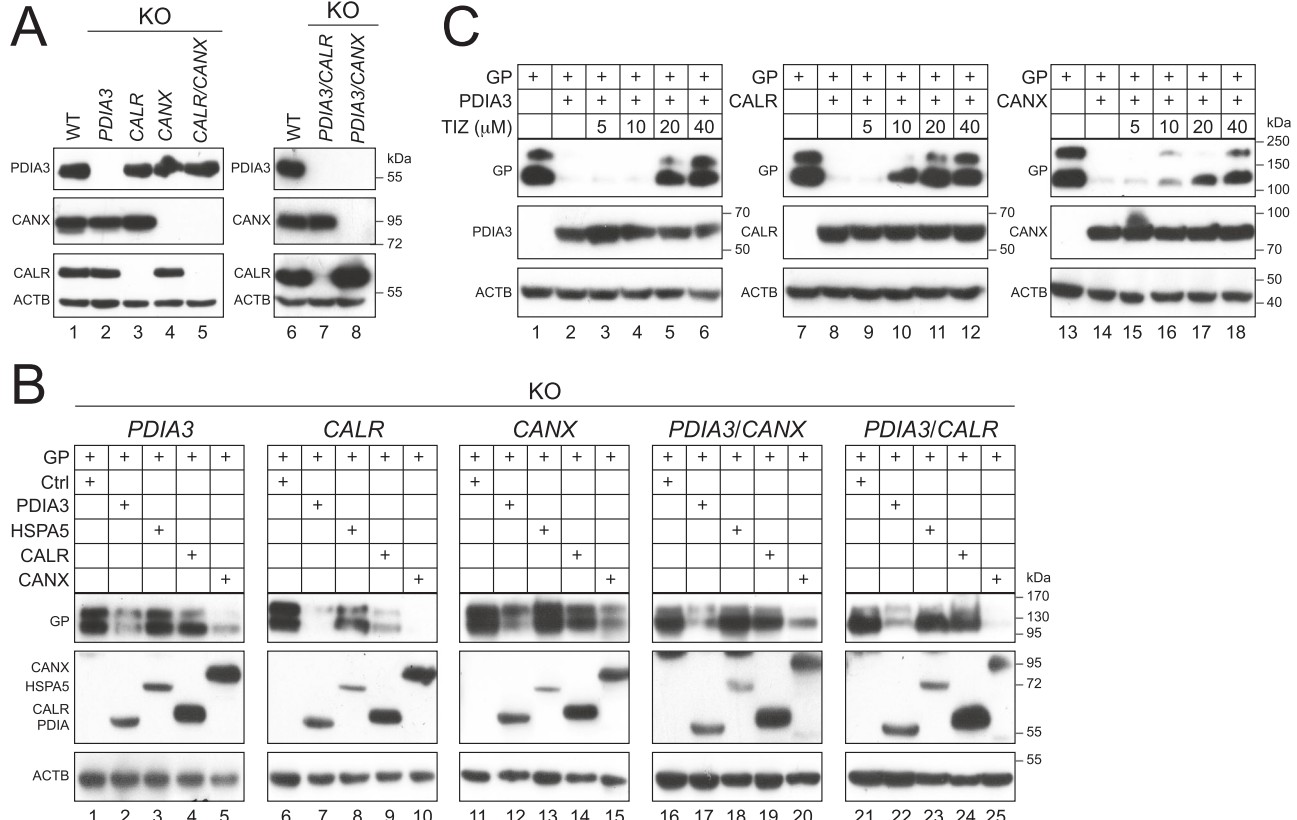

**Fig. 3 | Interrelationship between CANX, CALR, and PDIA3 in EBOV-GP$_{1,2}$ (GP) downregulation. A** Expression of PDIA3, CALR, and CANX was determined in HEK293T WT, single *PDIA3*-, *CALR*-, or *CANX*-KO, and dual *PDIA3/CALR*- or *PDIA3/CANX*-KO cells. Protein expression was determined by WB. PDIA3, CALR, and CANX were detected by their specific antibodies. **B** GP was expressed with indicated ER proteins in HEK293T WT and indicated KO cells. Protein expression was

detected by WB. GP was detected by anti-EBOV-GP; PDIA3 and CALR were detected by anti-Myc; HSPA5 was detected by anti-FLAG; CANX was detected by anti-HA. **C** GP was expressed with PDIA3, CALR, or CANX in HEK293T cells. Cells were treated with indicated amounts of Tizoxanide (TIZ), and protein expression was determined by WB. GP was detected by anti-EBOV-GP; PDIA3 and CALR were detected by anti-Myc; CANX was detected by anti-HA.

with EBOV-GP$_{1,2}$ that was suggested from the previous confocal microscopy data. In addition, they demonstrate that EBOV-GP$_{1,2}$ interacts with RNF26 via its TM3, TM4, and TM5 domains.

Finally, we tracked the RNF26 and EBOV-GP$_{1,2}$ interaction in live cells by bimolecular fluorescence complementation (BiFC). A basic yellow fluorescent protein Venus was divided into N-terminal (VN) and C-terminal (VC) fragments. HA-tagged VN and FLAG-tagged VC were fused to the C-terminus of RNF26 or EBOV-GP$_{1,2}$, respectively. Expression of RNF26-VN or GP-VC alone in cells did not produce any green fluorescence, while co-expression produced green fluorescence (Fig. 5G). This green fluorescence co-localized with the red

fluorescence produced from RNF26 after staining with fluorescent anti-HA, confirming the specificity of these BiFC signals. These BiFC signals also co-localized with CALR that has a C-terminal blue fluorescent protein (BFP) tag (Fig. 5G). Collectively, these results confirm RNF26 downregulation of and interaction with EBOV-GP$_{1,2.}$

### RNF26 supports CALR, but not CANX or PDIA3 to downregulate EBOV-GP$_{1,2}$

To understand whether RNF26 is required for CALR, CANX, and PDIA3 activities, we knocked out *RNF26* in HEK293T cells and generated three *RNF26*-KO cell lines 1-E4, 3-B8, and 3-F10 (Fig. 6A). When EBOV-GP$_{1,2}$

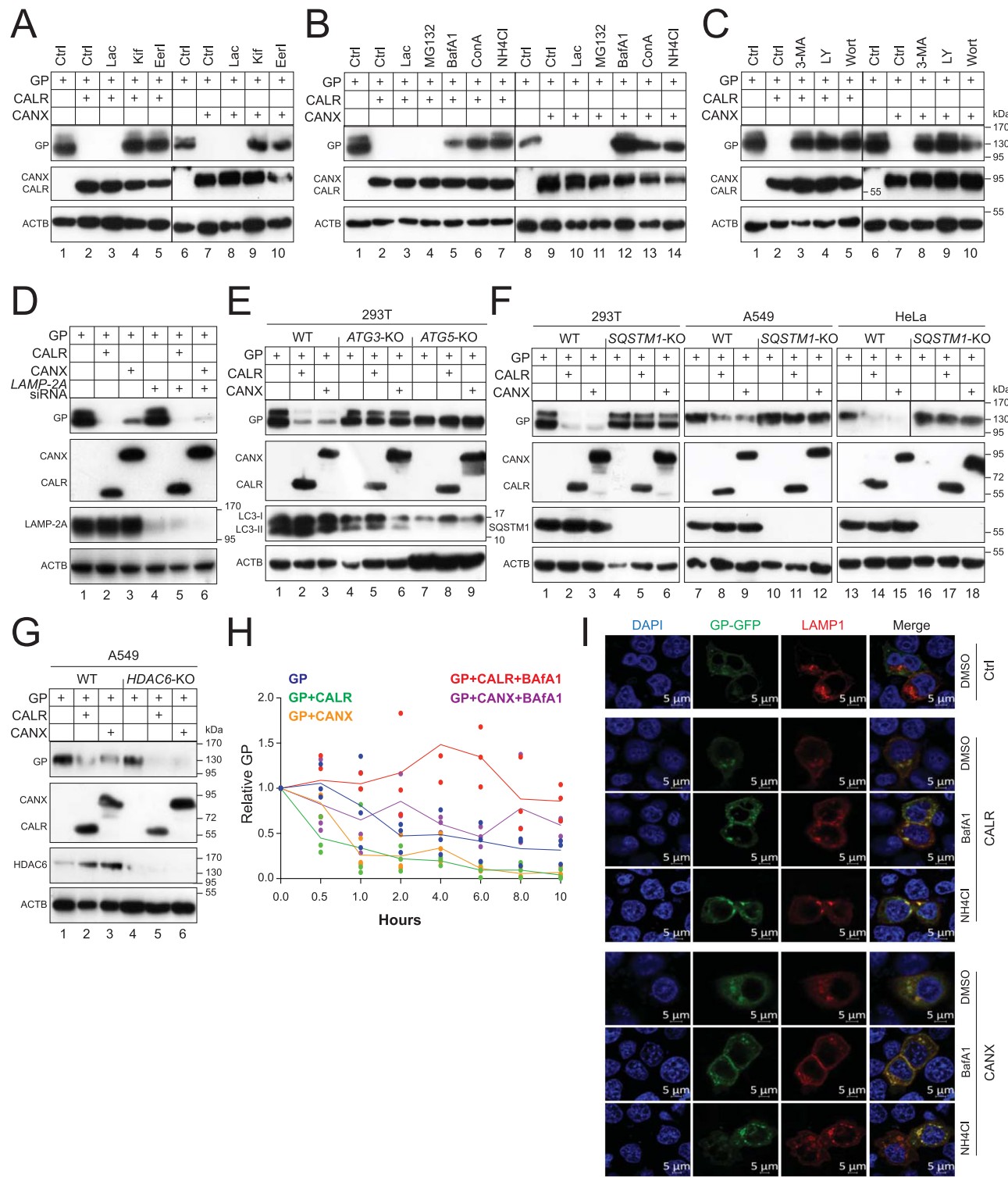

and RSTV-GP$_{1,2}$ were expressed in WT and 1-E4 cells, levels of their GP$_{1,2}$ expression were much higher in KO than WT cells (Fig. 6B). Thus, endogenous RNF26 also decreases GP$_{1,2}$ expression. Next, we detected PDIA3, CALR, and CANX activities in 1-E4 cells, using WT HEK293T cells as a control. PDIA3 and CANX downregulated EBOV-GP$_{1,2}$ expression in KO cells, whereas CALR did not (Fig. 6C). We then set up a co-IP assay to test whether RNF26 interacts with PDIA3, CALR, and CANX, using GFP as a control. We detected RNF26 interaction with CALR and CNAX, but not PDIA3 and GFP (Fig. 6D). Lastly, we used co-IP to compare EBOV-GP$_{1,2}$ polyubiquitination after expressing Ub in HEK293T cells in the presence of PDIA3, CALR, CANX, RNF26, and a catalytically inactive

mutant of RNF26 3 C/3 S. We detected polyubiquitinated GP$_{1,2}$ products at above 170 kDa, which were greatly increased by PDIA3, CALR, and CANX, but not RNF26, or its mutant 3 C/3 S (Fig. 6E, IP, bottom panel). Collectively, our results demonstrate that RNF26 is not responsible for EBOV-GP$_{1,2}$ polyubiquitination and only selectively supports CALR, but not CANX or PDIA3 activity toward GP$_{1,2}$.

## EBOV-GP$_{1,2}$ is polyubiquitinated with K27-linked Ub chains

To understand how EBOV-GP$_{1,2}$ becomes polyubiquitinated, we determined the Ub chain linkage on EBOV-GP$_{1,2}$. Seven Lys residues within Ub can be utilized for ubiquitination (K6, K11, K27, K29, K33,

**Fig. 4 | CALR and CANX target EBOV-GP$_{1,2}$ (GP) to the lysosomes for degradation via ERAD machinery. A** GP was expressed with CALR or CANX in HEK293T cells and treated with 20 μM lactacystin (Lac), 50 μM kifunensine (KIF), or 10 μM eeyarestatin I (EerI). DMSO was used as a vehicle control. Protein expression was detected by WB. GP was detected by anti-EBOV-GP; CALR and CANX were detected by anti-Myc or anti-HA. **B** GP was expressed with CALR or CANX in HEK293T cells and treated with 20 μM Lac, 20 μM MG132, 100 nM bafilomycin A1 (BafA1), 20 nM concanamycin A (ConA), or 20 mM NH$_4$Cl. DMSO was used as a vehicle control. Protein expression was detected by WB. GP was detected by anti-EBOV-GP; CALR and CANX were detected by anti-Myc or anti-HA. **C** GP was expressed with CALR or CANX in HEK293T cells and treated with 10 mM 3-methyladenine (3-MA), 20 μM LY294002 (LY), or 100 nM wortmannin (Wort). DMSO was used as a vehicle control (Ctrl). Protein expression was detected by WB. GP was detected by anti-EBOV-GP; CALR and CANX were detected by anti-Myc or anti-HA. **D** GP was expressed with CALR or CANX in HEK293T cells in the presence of *LAMP-2A*-siRNA. Protein expression was analyzed by WB. GP was detected by anti-EBOV-GP; CALR and CANX were detected by anti-Myc or anti-HA; LAMP-2A was detected by its specific antibody. **E** GP was expressed with CALR or CANX in HEK293T WT and *ATG3*- or *ATG5*-KO cells. Protein expression was analyzed by WB.

GP was detected by anti-EBOV-GP; CALR and CANX were detected by anti-Myc or anti-HA; LC3 was detected by its specific antibody. **F** GP was expressed with CALR or CANX in indicated WT and *SQSTM1*-KO cells. Protein expression was analyzed by WB. GP was detected by anti-EBOV-GP; CALR and CANX were detected by anti-Myc or anti-HA; SQSTM1 was detected by its specific antibody. **G** GP was expressed with CALR or CANX in A549 WT and *HDAC6*-KO cells. Protein expression was analyzed by WB. GP was detected by anti-EBOV-GP; CALR and CANX were detected by anti-Myc or anti-HA; HDAC6 was detected by its specific antibody. **H** GP was expressed with CALR or CANX in HEK293T cells. Cells were treated with 50 μM cycloheximide (CHX) in the presence or absence of 100 nM BafA1. Cells were collected at indicated time points and protein expression was detected by WB. Levels of GP expression were quantified from the blots by ImageJ and are presented as relative values, with the values from time 0 set to 1. Error bars represent SEMs calculated from three independent experiments. **I** GP with a C-terminal GFP tag (GP-GFP) was expressed with LAMP1 that has a dsRed tag in the presence or absence of CALR or CANX in HeLa cells. Cells with ectopic CALR or CANX were treated with DMSO, 100 nM BafA1, or 20 mM NH$_4$Cl. The co-localization of EBOV-GP with LAMP1 was determined by confocal microscopy. The scale bar denotes 5 μm.

K48, K63) and the site of linkage type directs the modified proteins to different cellular fates. Initially, we generated seven Ub mutants, K6R, K11R, K27R, K29R, K33R, K48R, and K63R, where each of its seven lysine residues were individually mutated to arginine. We also generated a Ub mutant, 7 K/R, in which all seven Lys residues were mutated to arginine. EBOV-GP$_{1,2}$ was expressed with each of these Ub mutants and PDIA3, CALR, or CANX in HEK293T cells, and EBOV-GP$_{1,2}$ polyubiquitination was again analyzed by co-IP. As we already observed, these three proteins promoted EBOV-GP$_{1,2}$ polyubiquitination in the presence of Ub; and importantly, similar levels of polyubiquitination were detected in the presence of mutants K6R, K11R, K29R, K33R, K48R, and K63R (Fig. 7A, IP, lower panels), but not K27R (lanes 6, 17, 28) and 7K/R (lanes 11, 22, 33). These results suggested that Ub K27 plays an important role in EBOV-GP$_{1,2}$ polyubiquitination.

To confirm the important role of K27 in EBOV-GP$_{1,2}$ polyubiquitination, we created another seven Ub mutants, K6, K11, K27, K29, K33, K48, and K63, that only express each of its seven Lys residues individually. When these mutants were used to analyze EBOV-GP$_{1,2}$ polyubiquitination, PDIA3, CALR, and CANX promoted EBOV-GP$_{1,2}$ polyubiquitination only in the presence of Ub and its mutant that only retains K27 (Fig. 7B, IP, low panels, lanes 6, 16, 26), but not mutants that retain K6, K11, K29, K33, K48, or K63. These results confirm the important role of the Ub K27 residue in EBOV-GP$_{1,2}$ polyubiquitination, suggesting that EBOV-GP$_{1,2}$ is polyubiquitinated via K27-linked Ub chains.

## RNF185 is responsible for EBOV-GP$_{1,2}$ polyubiquitination and degradation

K27-linked ubiquitination plays an important role in the innate immune response and T cell signaling[37]. Many E3 Ub ligases have been reported to catalyze K27-linked ubiquitination, including RNF185[38], membrane-associated RING-CH-type 8 (MARCH8)[39], and TRIM25[40]. To identify the critical E3 ligase in EBOV-GP$_{1,2}$ degradation, EBOV-GP$_{1,2}$ and Ub were expressed with RNF185, RNF26, TRIM25, or MARCH8, and their catalytically inactive mutants 3C/3A, 3C/3S, 2E/2A, or W114A in HEK293T cells. GP$_{1,2}$ proteins were pulled down and GP$_{1,2}$ polyubiquitination was detected, as we did previously. Among these four E3 ligases, RNF185, TRIM25, and MARCH8 strongly promoted EBOV-GP$_{1,2}$ polyubiquitination, whereas their mutants did not, and, as we already observed, neither did RNF26 and its mutant (Fig. 8A, IP, lower panel). In addition. TRIM25 and MARCH8 did not decrease GP$_{1,2}$ expression (Fig. 8B, lanes 7, 8). Nonetheless, MARCH8 inhibited GP$_{1,2}$ proteolytic cleavage, as we previously reported[26]. Notably, RNF185 decreased GP$_{1,2}$ expression as did PDIA3, CALR, CANX, and RNF28 (Fig. 8B), whereas its mutant 3 C/3 A did not (Fig.8A, Input, top panel, lanes 2, 3). Furthermore, GP$_{1,2}$ downregulation by RNF185 was blocked

by BafA1, ConA, 3-MA, Ly, and Wort, but not by Lac and MG132 (Fig. 8C, lanes 1–9). In contrast, GP$_{1,2}$ downregulation by RNF26 was only slightly blocked by BafA1, ConA, and 3-MA, but not any other inhibitors (Fig. 8C, lanes 10–18).

RNF185 is another E3 Ub ligase on the ER membrane that plays an important role in ERAD[41]. Its 192 aa comprise one N-terminal RING-finger and two C-terminal TM domains (Fig. 8D). To understand how EBOV-GP$_{1,2}$ is targeted, we created seven RNF185 deletion mutants, including Δ2–18, Δ19–38, Δ39–80, Δ81–132, Δ133–155, Δ171–192, and Δ133–192, that lack the indicated regions (Fig. 8D). In addition, we created four point-mutation mutants, C39A, C42A, C79A, and 3C/3A, that had the critical Cys residues in the RING-finger replaced with Ala (Fig. 8D). When RNF185 WT and these mutants were expressed with EBOV-GP$_{1,2}$ in HEK293T cells, all these RNF26 proteins were detected by WB (Fig. 8E). Mutants Δ39–80, Δ171–192, Δ133–192, and 3 C/3 A did not decrease GP$_{1,2}$ expression (Fig. 8E). These results demonstrate that the RING-finger and TM2 domain, and, importantly, the E3 Ub ligase activity are all required for RNF185 downregulation of GP$_{1,2}$ expression.

Next, we tested the RNF185 interaction with EBOV-GP$_{1,2}$ by co-IP. EBOV-GP$_{1,2}$ was expressed with GFP, RNF185 WT, and its mutants Δ2–18, Δ19–38, Δ39–80, Δ81–132, Δ133–155, Δ171–192, or Δ133–192 in HEK293T cells. GFP and RNF185 proteins were pulled down and their interactions with EBOV-GP$_{1,2}$ were determined. RNF185 WT and its mutant Δ133–155 pulled down EBOV-GP$_{1,2}$ with a similar efficiency; mutants Δ2–18, Δ19–38, and Δ81–132 pulled down EBOV-GP$_{1,2}$ with much less efficiency; GFP and mutants Δ171–192 and Δ133–192 did not pull down any EBOV-GP$_{1,2}$ (Fig. 8F). These results demonstrate that EBOV-GP$_{1,2}$ interacts with RNF185 via its TM2 domain. In addition, the 2–38 and 81–132 region that are amino terminal to the RING-finger or TM1 domain are also required for RNF185 interaction with EBOV-GP$_{1,2}$, likely by contributing to RNF185 structural integrity. To further validate these results, we tested whether these deletions affect RNF185 polyubiquitination activity toward EBOV-GP$_{1,2}$. Mutants Δ39–80, Δ81–132, Δ171–192, and Δ133–192 all showed a deficiency in promoting EBOV-GP$_{1,2}$ polyubiquitination, as did mutant 3C/3A when compared to the WT protein (Fig. 8G), consistent with our previous results.

## RNF185 polyubiquitinates EBOV-GP$_{1,2}$ on K673 in its cytoplasmic tail

EBOV-GP$_{1,2}$ has a very short cytoplasmic tail (CT) that only has four residues $^{673}$KFVF$^{676}$. To determine whether K673 is targeted for polyubiquitination, we created two EBOV-GP$_{1,2}$ mutants by replacing K673 with Ala (K673A) or deleting all these four residues (ΔCT). We found that RNF185, TRIM25, and MARCH8 could no longer

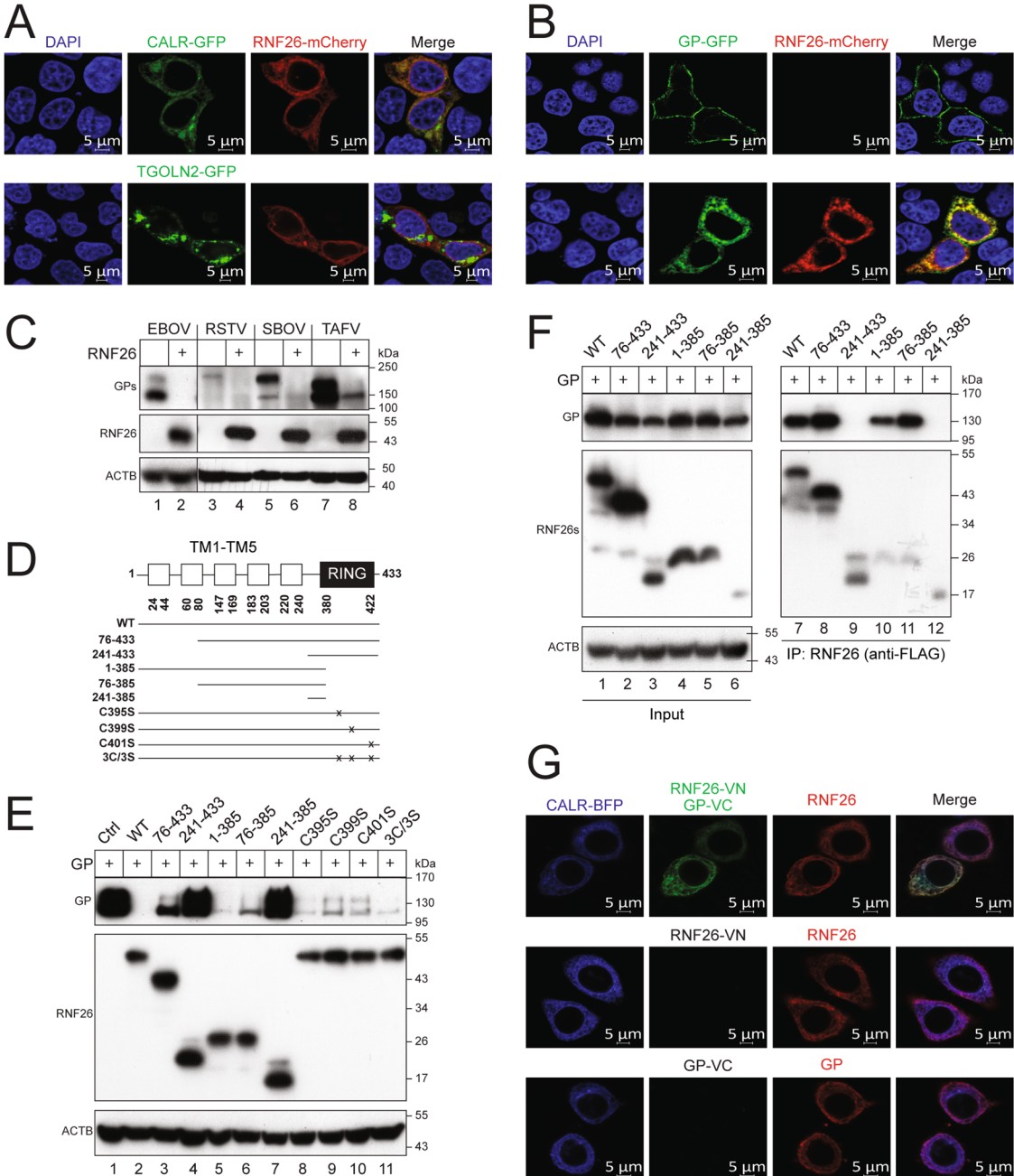

**Fig. 5 | RNF26 decreases EBOV-GP$_{1,2}$ expression in an E3 ubiquitin ligase activity-independent manner. A** RNF26 with a C-terminal mCherry tag was expressed with CALR or TGOLN2 (Trans-Golgi network integral membrane protein 2) that has a GFP-tag in HeLa cells. The RNF26 subcellular localization was determined by confocal microscopy. **B** EBOV-GP$_{1,2}$ (GP) with a C-terminal GFP tag was expressed with RNF26-mCherry in HeLa cells, and their co-localization was determined by confocal microscopy. **C** GP$_{1,2}$ proteins from indicated ebolaviruses were expressed with RNF26 in HEK293T cells. Protein expression was detected by WB. GPs were detected by anti-EBOV-GP; RNF26 was detected by anti-FLAG. **D** A schematic diagram of RNF26 is presented on the top. Five transmembrane (TM1–TM5) domains and a RING-finger are shown. Five RNF26 deletion mutants that target indicated regions and four point-mutation mutants that target each of C395, C399,

or C401 are indicated. **E** EBOV-GP$_{1,2}$ (GP) was expressed with RNF26 WT and its mutants in HEK293T cells. Protein expression was determined by WB. GP was detected by anti-EBOV-GP; RNF26 proteins were detected by anti-FLAG. **F** EBOV-GP$_{1,2}$ (GP) with a HiBiT tag was expressed with RNF26 WT and its mutants in HEK293T cells. RNF26 proteins were immunoprecipitated and their interactions with GP was analyzed by WB. GP was detected by anti-EBOV-GP; RNF26 proteins were detected by anti-FLAG; GFP was detected by its specific antibody. **G** RNF26-VN and EBOV-GP$_{1,2}$-VC were expressed with CALR that has a C-terminal Blue Fluorescent Protein (BFP) tag in HeLa cells. RNF26 was stained with a red-fluorescent antibody. The subcellular localization of the RNF26-GP complex was determined by confocal microscopy. The scale bar in **A**, **B**, and **G** denotes 5 µm.

polyubiquitinate mutants K673A and ΔCT (Fig. 9A, IP, lower panel). Thus, K673 is the polyubiquitination site on EBOV-GP$_{1,2}$ for all these three E3 Ub ligases. Consistently, we also found that PDIA3, CALR, and CANX did not promote the polyubiquitination of mutants K673A and ΔCT (Fig. 9B, IP, lower panel). Finally, we determined whether these two mutants are still sensitive to those E3 ligases and ER

proteins. We found that they were both resistant to MARCH8, TRIM25, RNF185, PDIA3, CALR, and CANX, although they were still as sensitive to RNF26 as was WT GP$_{1,2}$ (Fig. 9C). To further confirm the role of RNF185 in EBOV-GP$_{1,2}$ downregulation, we used siRNAs to knock down *RNF185* (Fig. 9D). Notably, *RNF185*-knockdown strongly increased GP$_{1,2}$ expression (Fig. 9E). Importantly, it also disrupted

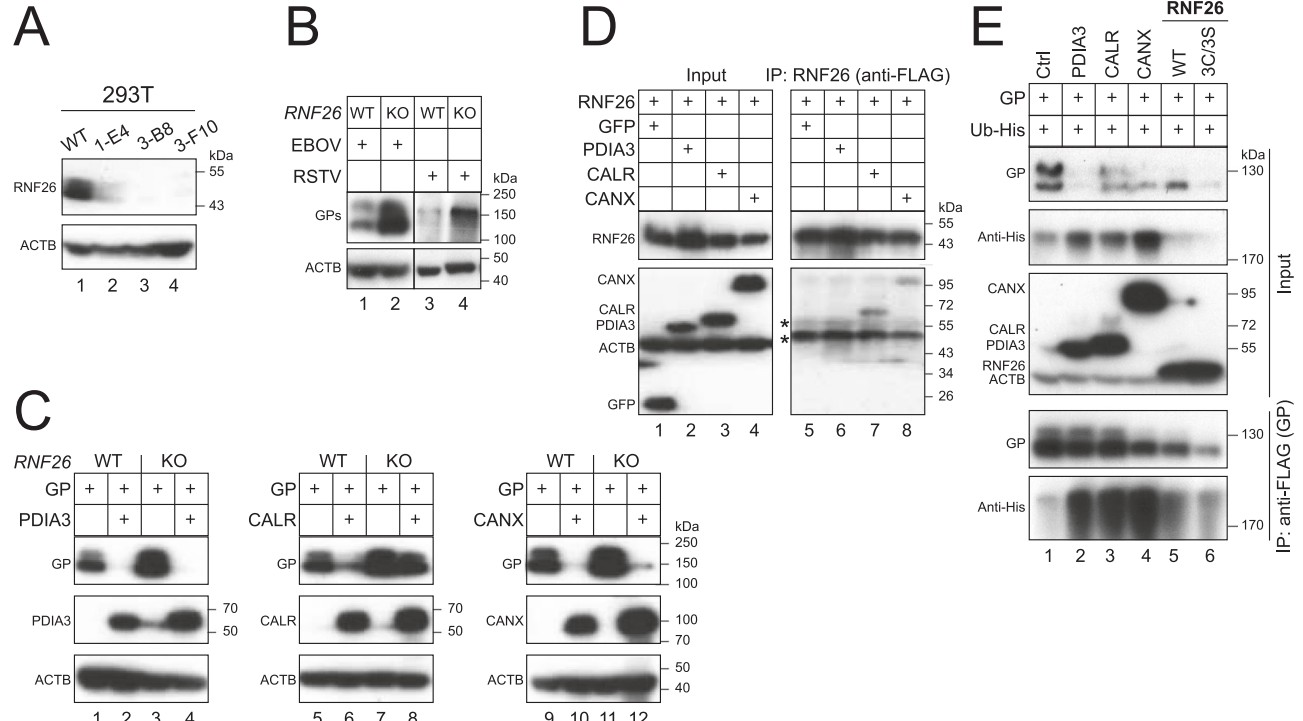

**Fig. 6 | RNF26 selectively supports CALR, but not CANX or PDIA3. A** Three *RNF26*-KO cell lines (1-E4, 3-B8, 3-F10) were generated from HEK293T cells using CRISPR/Cas9. RNF26 expression in these cells was determined by WB using its specific antibody. **B** EBOV and RSTV GP$_{1,2}$ proteins were expressed in HEK293T WT and *RNF26*-KO (1-E4) cells. GP expression was detected by WB using anti-EBOV-GP. **C** EBOV-GP$_{1,2}$ (GP) was expressed with PDIA3, CALR, or CANX in HEK293T WT and *RNF26*-KO (1-E4) cells. Protein expression was detected by WB. GP was detected by anti-EBOV-GP; PDIA3 and CALR were detected by anti-Myc; CANX was detected by anti-HA. **D** RNF26 was expressed with GFP, PDIA3, CALR, or CANX in HEK293T cells. RNF26 was co-immunoprecipitated (IP) from cell lysate (Input) and its interactions with these proteins was determined by WB. GP was detected by anti-EBOV-GP; PDIA3 and CALR were detected by anti-Myc; CANX was detected by anti-HA; RNF26 was detected by anti-FLAG; GFP was detected by its specific antibody. **E** GP was expressed with His-tagged ubiquitin (Ub) and PDIA3, CALR, CANX, and RNF26 or its catalytically inactive mutant (3C/3S) in HEK293T cells. GP proteins were immunoprecipitated and GP polyubiquitination was determined by WB. GP was detected by anti-EBOV-GP; PDIA3 and CALR were detected by anti-Myc; CANX was detected by anti-HA; Ub was detected by anti-His.

GP$_{1,2}$ downregulation by PDIA3, CALR, and CANX (Fig. 9F). Collectively, these results demonstrate that calnexin cycle co-opts RNF185 to polyubiquitinate EBOV-GP$_{1,2}$ on K673, resulting in EBOV-GP$_{1,2}$ degradation via reticulophagy.

## Discussion

Responses to viral infections are expected to have played an important role in the evolution of mammalian proteostasis network and therefore provide a unique system to understand its functional mechanism. To this point, all known clients for reticulophagy have been cellular proteins[42]. We recently reported that Ebolavirus GP$_{1,2}$ proteins are highly vulnerable to PDIA3, even at physiological levels that more generally promote oxidative protein folding[24]. We now show that the calnexin cycle indeed triggers the GP$_{1,2}$ misfolding via PDIA3, and that misfolded GP$_{1,2}$ proteins are targeted by ERAD machinery to lysosomes for degradation. Thus, Ebolaviruses utilize this cellular pathway to downregulate GP$_{1,2}$ expression and promote fitness. Importantly, Ebolavirus GP$_{1,2}$ can be used to study the mechanism of reticulophagy and interactions in the ER proteostasis-network that are still poorly understood.

Reticulophagy is executed through three pathways including macro-ER-phagy, micro-ER-phagy, and vesicular delivery. Macro-ER-phagy requires both the LC3 lipidation machinery and the autophagosome biogenesis machinery, and the others may only require the LC3 lipidation machinery or neither of them[4]. GP$_{1,2}$ degradation by the calnexin cycle was blocked by 3-Methyladenine, Wortmann, and LY294002 that inhibit autophagosome formation by targeting the PI3K-III complex, and it was also blocked by knocking out *ATG-3* and *ATG-5* that are required for LC3-I lipidation. Thus, Ebolavirus GP$_{1,2}$

proteins are degraded by the macro-ER-phagy pathway. An important question in this pathway is how misfolded proteins are segregated into an ER subdomain that is captured by autophagosomes. We reported that EBOV-GP$_{1,2}$ proteins are targeted by ER-resident class I α-mannosidases, including MAN1B1/ERManI (mannosidase alpha class 1B member 1), EDEM1 (ER degradation enhancing α-mannosidase like 1), and EDEM2[24]. The GP$_{1,2}$ degradation was blocked by kifunesine that specifically inhibits class I α-mannosidases and eeyarestatin I that specifically inhibits VCP/p97. It has previously been thought that class I α-mannosidases extensively cleave α1,2-linked mannose residues on misfolded glycoproteins in the ER, which are then translocated to the cytoplasm by the action of VCP/p97 via ATP hydrolysis for exclusive elimination by proteasomes during ERAD. Our results demonstrate that these essential ERAD components can also play a role in promoting reticulophagy. We speculate that VCP/p97 is involved in the formation of ER subdomains, where misfolded client glycoproteins are recruited after being demannosylated by class I α-mannosidases for reticulophagy.

Polyubiquitination plays an important role in cargo recognition during mitophagy. When mitochondria are damaged, the mitochondrial kinase PINK1 is phosphorylated to recruit the E3 Ub ligase Parkin, which in turn polyubiquitinates proteins on the mitochondrial outer membrane. Ubiquitination leads to the recruitment of soluble Sequestosome-1-like receptors (SLRs), such as SQSTM1/p62 that has a ubiquitin-associated (UBA) domain and a LC3-interacting region (LIR), and then subsequent mitophagy[43]. We showed that the GP$_{1,2}$ degradation was blocked by knocking out *SQSTM1/p62*. In addition, it was reported that TRIM13 self-polyubiquitination is required for degradation of HSPA5 and CALR via reticulophagy[44].

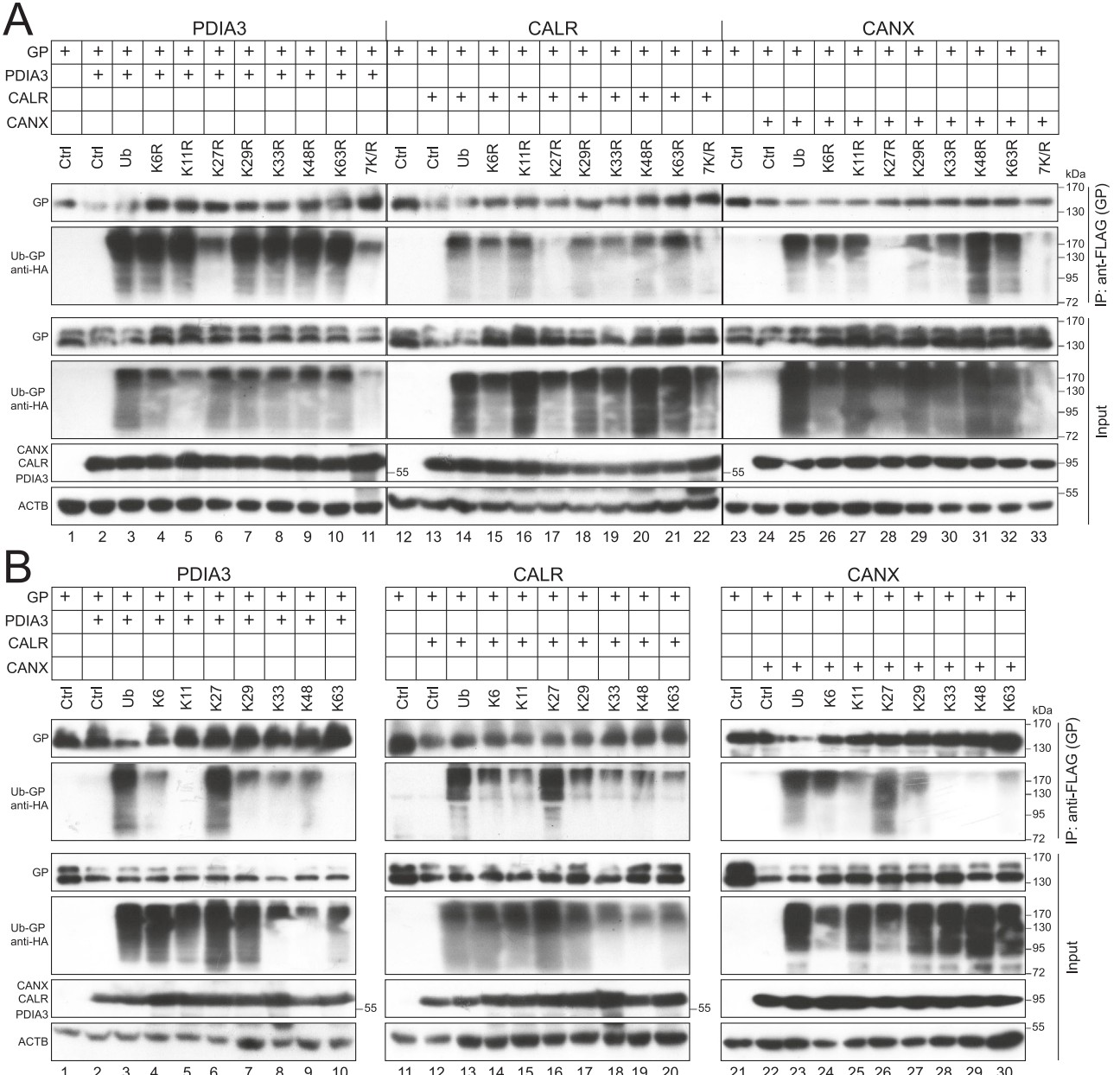

**Fig. 7 | EBOV-GP$_{1,2}$ (GP) is polyubiquitinated with K27-linked ubiquitin chains.**
**A** GP was expressed with Ub or its mutants bearing indicated single K-to-R mutation, and PDIA3, CALR, or CANX, in HEK293T cells. GP proteins were immunoprecipitated and GP polyubiquitination was analyzed by WB. **B** GP was expressed with Ub or its mutants expressing indicated single lysine residue, and PDIA3, CALR, or CANX, in HEK293T cells. GP proteins were immunoprecipitated and GP polyubiquitination was analyzed by WB. In **A** and **B**, GP was detected by anti-EBOV-GP; PDIA3, CALR, and CANX were detected by anti-Myc; Ub was detected by anti-HA.

These results encouraged us to search for the E3 Ub ligase for EBOV-GP$_{1,2}$ autophagic degradation. Our initial target was RNF26, which polyubiquitinates SQSTM1/p62 to liberate its UBA domain and recruit polyubiquitylated cargoes for selective autophagy[33,34]. Although we found that RNF26 alone could decrease EBOV-GP$_{1,2}$ expression and was required for EBOV-GP$_{1,2}$ downregulation by CALR, it did not polyubiquitinate EBOV-GP$_{1,2}$. As RNF26 activity was only modestly blocked by autolysosome inhibitors or proteasome inhibitors, and RNF26 plays an important role in retaining lysosomes, endosomes, and vesicles in the perinuclear region of the cell[34], we speculate that RNF26 may directly inhibit EBOV-GP$_{1,2}$ expression by blocking the intracellular trafficking events via its TM3, TM4, and TM5 domains.

We found that EBOV-GP$_{1,2}$ is polyubiquitinated on K673 via K27-linked ubiquitination, which is required for its downregulation.

Screening E3 Ub ligases that have such activity, we identified RNF185 as the E3 Ub ligase for EBOV-GP$_{1,2}$ degradation. RNF185 degrades membrane proteins, such as the cystic fibrosis transmembrane conductance regulator (CFTR), ADP-ribosylation factor-like protein 6-interacting protein 5 (ARL6IP5), and human Ergosterol biosynthesis protein (Erg) 11 homolog Cytochrome P450 (CYP) 51A1 by classical ERAD that employs proteasomes for substrate degradation[8,41,45]. In addition, RNF185 catalyzes K27-linked polyubiquitination of the cyclic GMP-AMP synthase (cGAS) and potentiates cGAS enzymatic activity that plays a key role in the innate immune response[38]. We found RNF185 not only polyubiquitinated EBOV-GP$_{1,2}$ on K673, but also decreased EBOV-GP$_{1,2}$ expression via autolysosomes. All these activities required its RING-finger and E3 Ub ligase activity. We also detected RNF185 interaction with EBOV-GP$_{1,2}$ by co-IP and mapped the interactive region to its TM1 and TM2 domains. Thus, RNF185 plays an

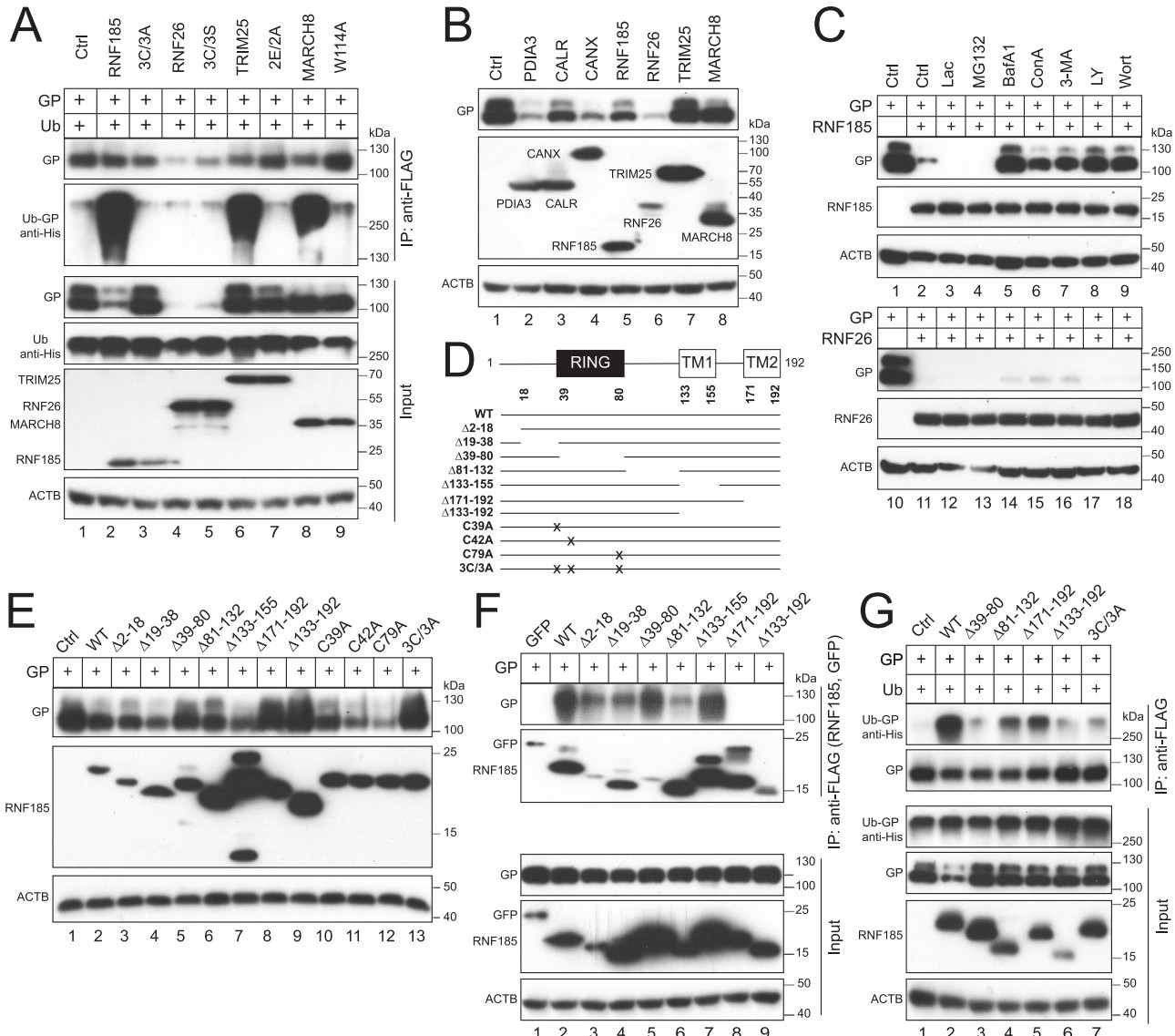

**Fig. 8 | RNF185 polyubiquitinates and degrades EBOV-GP$_{1,2}$ (GP). A** GP was expressed with His-tagged Ub in the presence of indicated E3 Ub ligases or their catalytically inactive mutants in HEK293T cells. GP and E3 proteins were immunoprecipitated and GP polyubiquitination was analyzed by WB. GP was detected by anti-EBOV-GP; RNF26, RNF185, and TRIM25 were detected by anti-FLAG; MARCH8 was detected by anti-HA; Ub was detected by anti-His. **B** GP was expressed with indicated proteins in HEK293T cells. Protein expression was determined by WB. GP, RNF26, RNF185, TRIM25, and MARCH8 were detected as in **A**; PDIA3 and CALR were detected by anti-Myc; CANX was detected by anti-HA. **C** GP was expressed with RNF185 or RNF26 and treated with indicated inhibitors as we did previously. Protein expression was determined by WB. GP was detected by anti-EBOV-GP; RNF26 and RNF185 proteins were detected by anti-FLAG. **D** A schematic diagram of RNF185 is

presented. Two transmembrane (TM1, TM2) domains and a RING-finger are shown. Indicated RNF26 mutants were constructed. **E** GP was expressed with RNF185 and its mutants in HEK293T cells. Protein expression was determined by WB. GP was detected by anti-EBOV-GP; RNF185 proteins were detected by anti-FLAG. **F** GP was expressed with GFP, RNF185, and indicated RNF185 mutants in HEK293T cells. GFP and RNF185 proteins were immunoprecipitated and their interaction with GP was analyzed by WB. GP was detected by anti-EBOV-GP; GFP and RNF185 proteins were detected by anti-FLAG. **G** GP was expressed with Ub with a His-tag and RNF185 or its indicated mutants in HEK293T cells. GP and RNF185 proteins were immunoprecipitated and GP polyubiquitination was determined by WB. GP was detected by anti-EBOV-GP; RNF185 proteins were detected by anti-FLAG; Ub was detected by anti-His.

important role in reticulophagy to degrade misfolded glycoproteins. Notably, although TRIM25 and MARCH8 polyubiquitinated EBOV-GP$_{1,2}$ similarly, they did not downregulate EBOV-GP$_{1,2}$. The mechanism for this surprising result remains to be elucidated. We previously reported that MARCH8 blocks EBOV-GP$_{1,2}$ proteolytic cleavage by furin in an E3 ligase activity-dependent manner[26]. MARCH8 also blocked this cleavage of EBOV-GP$_{1,2}$ mutants K673A and ΔCT (Fig. 9C, lanes 1–6). These results demonstrate that MARCH8 targets another cellular protein to inhibit furin activity.

As CALR could not act alone, CANX played a central role in EBOV-GP$_{1,2}$ downregulation by the calnexin cycle. CANX interacts with a reticulophagy receptor RETREG1 (reticulophagy regulator 1) and the

lipidated form of LC3 (LC3-II), and targets aberrant alpha1-antitrypsin Z (ATZ) and type I procollagen (PC1) to reticulophagy for degradation[46,47]. Notably, CALR and PDIA3 are also required for clearance of PC1, but not AZT. Thus, EBOV-GP$_{1,2}$ shares a similarity with PC1 in that CANX, CALR, and PDIA3 all play a role in the downregulation by protein degradation. Based on these observations, we propose that the CANX-RETREG1 axis plays a role in EBOV-GP$_{1,2}$ downregulation (Fig. 10). We suggest that the calnexin cycle co-opts RNF185 in the ER membrane to polyubiquitinate EBOV-GP$_{1,2}$ on K673 via K27-linked ubiquitination in GP$_{1,2}$ CT. Polyubiquitinated EBOV-GP$_{1,2}$ is then recruited into autophagosomes via SQSTM1/p62 and degraded in autolysosomes. The VCP/p97-ATPase may participate in this process

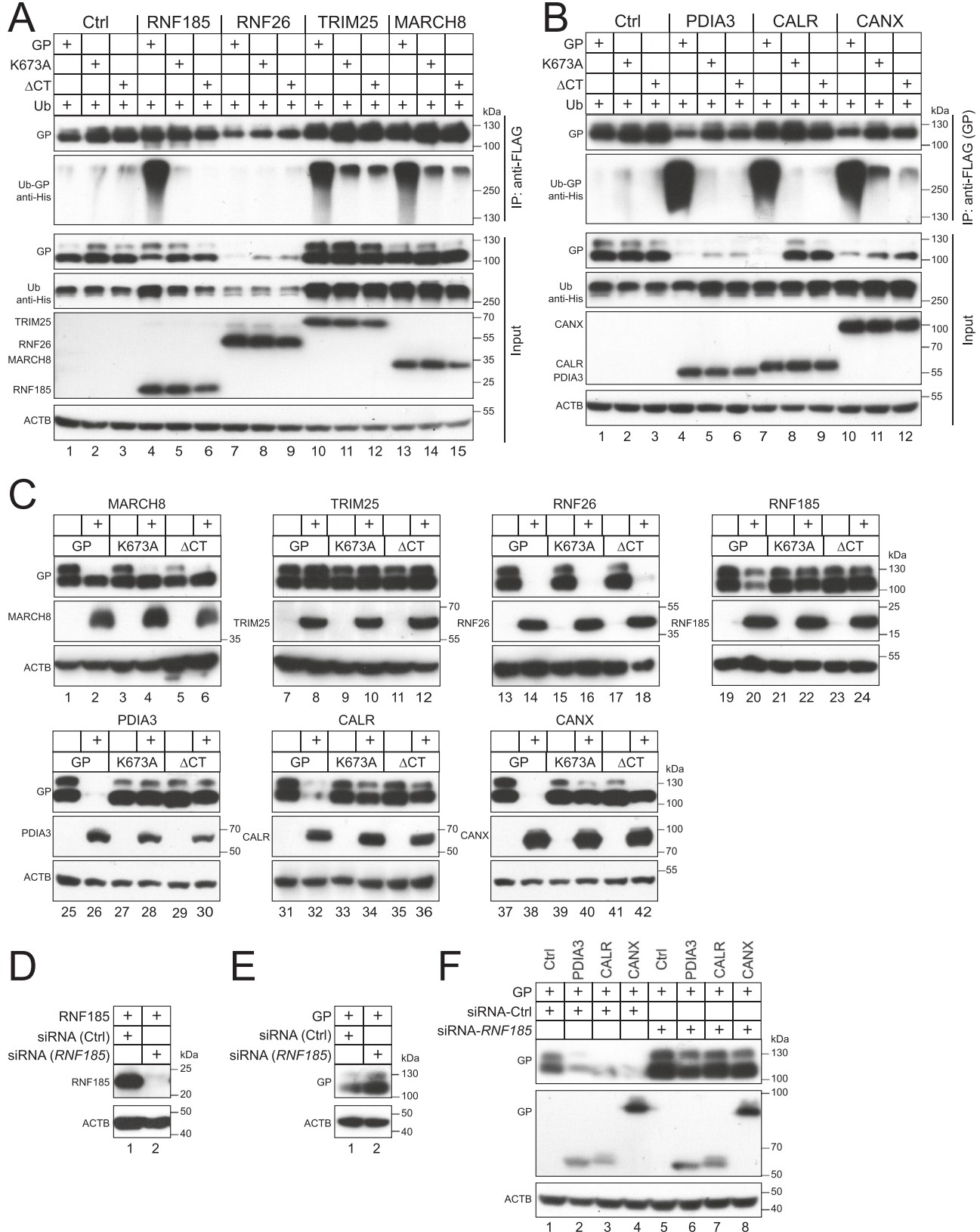

during invagination and pinching off from the ER subdomains, which are captured by autophagosomes.

The evolution of the proteostasis system in mammalian cells represents an overall cellular response to viral infections. Thus, the use of viruses to study proteostasis systems may identify unexpected linkages in the network that were previously considered to function independently. The mechanism for these surprising results remains to be elucidated. A full understanding of this novel cellular degradation pathway holds the potential for identification of new strategies for treating Ebola virus disease (EVD) by shutting down Ebolavirus GP$_{1,2}$ expression through enhanced degradation.

**Fig. 9 | RNF185 polyubiquitinates EBOV-GP$_{1,2}$ (GP) on K673 in the cytoplasmic tail. A** GP and its mutants K673A or ΔCT were expressed with indicated E3 Ub ligases in HEK293T cells. GP and E3 proteins were immunoprecipitated and GP polyubiquitination was analyzed by WB. GP was detected by anti-EBOV-GP; RNF26, RNF185, and TRIM25 were detected by anti-FLAG; MARCH8 was detected by anti-HA; Ub was detected by anti-His. **B** GP and its two mutants were expressed with His-tagged Ub in the presence of PDIA3, CALR, or CANX in HEK293T cells. GP proteins were immunoprecipitated and GP polyubiquitination was analyzed by WB. GP was detected by anti-EBOV-GP; PDIA3 and CALR were detected by anti-Myc; CANX was detected by anti-HA; Ub was detected by anti-His. **C** GP and its two mutants were expressed with indicated E3 ubiquitin ligases or ER proteins in HEK293T cells. Protein expression was determined by WB. GP was detected by anti-EBOV-GP; MARCH8 and CANX were detected by anti-HA; RNF26, RNF185, and TRIM25 were detected by anti-FLAG; PDIA3 and CALR were detected by anti-Myc. **D** RNF185 was expressed with a control (Ctrl) or *RNF185*-specific siRNAs in HEK293 cells. RNF185 expression was determined by WB using anti-FLAG. **E** GP was expressed with PDIA3, CALR, or CANX in the presence of Ctrl or *RNF185*-specific siRNAs in HEK293T cells. GP expression was determined by WB using anti-EBOV-GP. **F** GP was expressed with CALR, CANX, or PDIA3, and indicated siRNAs in HEK293T cells. GP expression was determined by WB using anti-EBOV-GP.

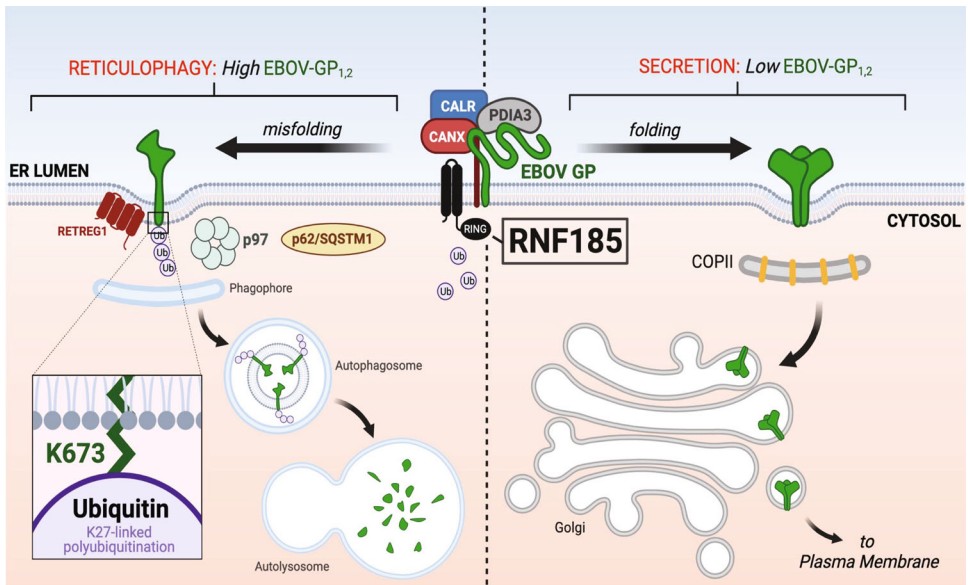

**Fig. 10 | A model of how EBOV-GP$_{1,2}$ expression is regulated in the ER via reticulorphagy.** When GP$_{1,2}$ levels are low in the ER, CANX-CALR cycle promotes GP$_{1,2}$ folding. Properly folded GP$_{1,2}$ is released from the ER to the Golgi via COPII vesicles and secreted to the plasma membrane. However, when GP$_{1,2}$ levels become high in the ER, the same CANX-CALR cycle starts to trigger GP$_{1,2}$ misfolding. RNF185 polyubiquitinates aberrant GP$_{1,2}$ on K673 in the cytoplasmic tail via K27-linked ubiquitination. Polyubiquitinated GP$_{1,2}$ is recruited to the phagophore assembly site by RETREG1 and packaged into autophagosomes via p97 ATPases and p62/SQSTM1, resulting in GP$_{1,2}$ degradation in autolysosomes. Thus, CANX-CALR cycle promotes ebolavirus fitness in host cells by regulating GP$_{1,2}$ expression levels during infection.

## Methods

### Cell lines

Human embryonic kidney (HEK) 293 cell line transformed with SV40 large T antigen (293 T), human cervical carcinoma cell line HeLa, African green monkey kidney epithelial cell line Vero E6, human lung carcinoma cell line A549, and human hepatoblastoma cell line Hep G2 were purchased (ATCC CRL-3216, CRM-CCL-2, CRL-1586, CRM-CCL-185, or HB-8065). Human hepatoma cell line Huh7 was from National Virus Resource Center, Wuhan Institute of Virology, Chinese Academy of Sciences. HEK293T *CALR*-KO, *CANX*-KO, and *CANX/CALR*-KO cell lines were reported[25]. HEK293T *PDIA3*-KO, *ATG3*-KO, *ATG5*-KO, and *SQSTM1/p62*-KO cell lines, HeLa *SQSTM1/p62*-KO, and A549 *SQSQM1/p62*-KO, and *HDAC6*-KO cell lines were reported[24]. HEK293T *RNF26*-KO, *PDIA3/CALR*-KO, and *PDIA3/CANX*-KO cell lines were generated by CRISPR/Cas9. All these cells were maintained in Dulbecco's modified Eagle medium (DMEM) (Thermo Fisher Scientific, 11965092) supplemented with 10% fetal bovine serum and 1% penicillin-streptomycin (pen-strep) (Thermo Fisher Scientific, 10378016), and cultivated at 37 °C in humidified atmosphere in a 5% $CO_2$ incubator.

To differentiate THP1 into macrophages (THP1-MΦ), human monocytic THP1 cells (ATCC, TIB-202) were grown in RPMI 1640 medium (Thermo Fisher Scientific, A4192301) supplemented with 10% FBS and 1% pen-strep in 6-well plates ($1.5 \times 10^6$ cells per well) and treated with 100 nM phorbol 12-myristate 13-acetate (PMA; Sigma-Aldrich, P8139) for 2 days. Primary peritoneal macrophages (MΦ) were isolated from mouse peritoneum and cultured in RPMI 1640 supplemented with 10% FBS and 1% pen-strep.

### CRISPR-Cas9 knockout (KO) and siRNA knockdown

To create *PDIA3/CALR*-KO and *PDIA3/CANX*-KO cell lines, a DNA fragment that contained the U6 promoter, a 19-bp target sequence specific for *PDIA3*, *CALR*, or *CANX*, a small guide (sg) RNA scaffold, and a U6 termination signal sequence was synthesized and subcloned into the pGEM-T Easy vector (Promega). Two pairs of oligos were designed for *PDIA3*: 5′-TGTACAAAAAAGCAGGCTT-3′/5′-GGATAAACTTTTGTGTTTC GTCCTTTCCAC-3′ and 5′-AATTAAAAAGTTTATCCGTTTTAGAGCTA G-3′/5′-TAATGCCAACTTTGTACAAGAAAGCT-3′. One pair of oligos was designed for *CALR*: 5′-GGCAAGTTCTACGGTGACG-3′/5′-CGTCACC GTAGAACTTGCC-3′. One pair of oligos was designed for *CANX*: 5′-G GGTGGATTTTATCCAAAGCC-3′/5′- TCACCACAAGATTGGTCAACC-3′. The vector was transfected with a human codon-optimized Cas9 expression vector and pEGFP-C1 into HEK293T cells. Twenty-four hours later, GFP-positive cells were isolated by fluorescence-activated cell sorting (FACS) and subjected to cloning by limiting dilution. After 10–14 days, knockout clones were screened by WB.

To create *RNF26*-KO cell line, DNA oligos expressing its sgRNAs were designed and directly cloned into the pSpCas9 (BB)−2A-GFP (PX458) vector after *BbsI*-digestion, which was obtained from Feng

Zhang through Addgene (48138). Two pairs of oligos were designed to target *RNF26*: 5'-caccAGTCTTCTGCACTTGGGCCG-3'/5'-aaacCGGCC-CAAGTGCAGAAGACTc-3' and 5'-caccGTTGGACCTCAACTTCCTGC-3'/5'-aaacGCAGGAAGTTGAGGTCCAACc-3'. After transfection of HEK 293T cells with these vectors, single GFP-positive cells were sorted out by FACS, and knockout clones were screened by WB.

*LAMP2A* and *RNF185* were knocked down by siRNAs using RNA oligo pairs 5'-GCACCCAUGCUGGAUAUUU-3'/5'-AUAUCCAGCAUGAU GGUGCUU-3', or 5'-GUGGCUUCCAGAUGUCUUUU-3'/5'-AAAGACAUC UGGAAGCCACUU-3'. HEK293T cells were transfected with these siR-NAs, and after 24 h, their expression was determined by WB.

### Expression vectors

pcDNA3.1-EBOV-GP, pcDNA3.1-EBOV-GPΔMLD, pCMV3-EBOV-VP40, pNL-Luc-△Env, pMSCVneo-CANX-HA, pCMV6-CALR-Myc, and pCAGGS-CANX-HA were reported[25]. pcDNA3.1-GFP-FLAG-HA, pRK5-Ub-His, pCAGGS-PDIA3-Myc, pCAGGS-PDIA3-HA, pcDNA3.1 vectors expressing EBOV-GP, GP△MLD, sGP or ssGP with a N-terminal HiBiT tag, pCAGGS-MARV-GP with a C-terminal HiBiT tag, pcDNA3.1 vectors expressing GPs from BDBV, RETV, SBOV, or TAFV with a HA-FLAG-tag, and vectors expressing HIV-1 Env, IAV-H5 HA, and VSV-G were reported[24]. pCAGGS-MARCH8-HA and its W114A mutant were reported[26]. pEGFPC1 was from Takara.

pCMV6-CANX-Myc-FLAG was constructed from pCAGGS-CANX-HA by PCR and ASiSI/MluI digestion. pCAGGS vectors expressing MERS-CoV-S, SRAS-CoV-S, and SARS-CoV2-S with a FLAG-tag were constructed by PCR and EcoRI/XhoI digestion. pEGFPN1 vectors expressing CALR, TGLON2, and EBOV-GP were constructed by PCR and XhoI/BspEI digestion. pcDNA3.1-EBOV-GP-HA was constructed by PCR and NheI/HindIII digestion. pmBFPN1-CALR was constructed from pEGFPN1-CALR by replacing *EGFP* with *mBFP* via AgeI/NotI digestion. pRK5-HA-Ub-WT was obtained from Ted Dawson via Addgene (#17608). pCMV6-RNF26-Myc-FLAG, pCMV6-RNF185-Myc-FLAG, and pcDNA3.1-TRIM25-FLAG were ordered from CoME, ORIGENE, or JIN-SIRUI. pCAGGS-RNF26-mCherry was constructed from pCAGGS-ACE2-mCherry by PCR and EcoRI digestion followed by homologous recombination (Vazyme, C113-01). pcDNA3.1-RNF26-VN-HA was constructed from pcDNA3.1-ACE2-VN-HA by PCR and XhoI/BspEI digestion. pcDNA3.1-FLAG-EBOV-GP-FLAG-VC was constructed from pcDNA3.1-ACE2-FLAG-VC by PCR and XhoI/BspEI digestion. RNF26 and RNF185 deletion mutants were created by PCR and ASiSI/MluI digestion followed by homologous recombination. pcDNA3.1-FLAG-EBOV-sGP, pcDNA3.1-FLAG-EBOV-ssGP, pcDNA3.1-EBOV-GP$_{1,2}$ K673A and ΔCT mutants, pRK5-HA-Ub mutants, and pCMV6-RNF26, pCMV6-RNF185, and pcDNA3.1-TRIM25 catalytically inactive mutants were constructed by using the QuikChange site-directed mutagenesis kit (OMEGA BIO-TEK, D6492-01). All vectors were sequenced by Sanger DNA sequencing to confirm their inserts. Vector maps were created with SnapGene 6.0.2. Detailed experimental procedures and primer sequences for construction of these vectors are available upon request. Plasmids were prepared using Maxiprep kits (TIANGEN Bio-tech, DP117).

### Transfection

293T cells were transfected using linear 25 kDa polyethyleneimine (PEI; Polysciences, 23966-2). Briefly, to transfect cells in a 10-cm culture dish with 70% confluency, 9 μg plasmid DNAs were mixed with 27 μg PEI (stock solution at 1 mg/mL) and diluted into 500 μL Opti-MEM (Thermo Fisher Scientific, 31985062). After a 15-min incubation at room temperature, DNA-PEI complexes were added into the cell culture. After culturing cells for 4 to 6 h, cells were washed with PBS and fresh media was added for culturing another 42 h before samples were harvested. HeLa and A549 cells were transfected using Lipofectamine 3000 according to the manufacturer's protocol (Thermo Fisher Scientific, L3000).

### Preparation of EBOV virus-like particles

EBOV VLPs were generated as previously[24]. Briefly, 293T cells were transfected with EBOV-GP and EBOV-VP40 expression vectors in the presence or absence of ectopic PDIA3, CALR or CANX expression. After 48 h of transfection, the culture supernatants were collected, clarified by low-speed centrifugation, and passed through a 0.45-μm syringe filter to remove any cell debris. To analyze the levels of EBOV GP incorporation, supernatants containing EBOV VLPs were centrifuged at 120,000 x *g* at 4 °C for 2 h using a Beckman SW-32Ti rotor.

### Preparation of HIV-1 pseudoviruses for entry analysis

Lentiviral pseudo-virions expressing EBOV GP were generated as we did previously[25]. Briefly, 293T cells were transfected with pNL-Luc-△Env and a GP-expression vector in the presence or absence of ectopic PDIA3, CALR or CANX expression. After 48 h of transfection, pseudo-virion-containing culture supernatants were collected, clarified by low-speed centrifugation, and passed through a 0.45-μm syringe filter to remove any cell debris. After quantifying by a HIV-1 p24$^{Gag}$ ELISA kit (Novus, NBP2-79359), an equal number of pseudo-virions were used to infect 293T cells. After 48 h, luciferase activities were determined using Bright-Glo Luciferase Assay kit (Promega, N1130).

### Production of EBOV trVLPs and infection

Experiments were performed as reported previously[27]. EBOV trVLP vectors including p4cis-vRNA-RLuc, pCAGGS-L, pCAGGS-NP, pCAGGS-VP30, pCAGGS-VP35, pCAGGS-Tim1, and pCAGGS-T7 were provided by Heinz Feldmann (NIH). Briefly, to produce p0 trVLPs, HEK293T cells were seeded in 6-well plates at 4 ×10$^5$ per well in 2 mL medium and cultured for 24 h. EBOV trVLP system plasmids were diluted from stocks into 125 μL Opti-MEM per well that included 125 ng pCAGGS-NP, 125 ng pCAGGS-VP35, 75 ng pCAGGS-VP30, 1000 ng pCAGGS-L, 250 ng p4cis-vRNA-Rluc, and 250 ng pCAGGS-T7. The plasmids were combined with 7.5 μL TransIT®-LT1 Transfection Reagent (Mirus Bio, 2305). After incubation at room temperature for 15 min to promote complex formation, the transfection mixes were gently added into cell culture. After 24 h of transfection, the media was exchanged for 4 mL growth media with 5% FBS, and further incubated for 72 h. All supernatants from p0 producer cells were collected followed by centrifugation at 3000 x *g* for 5 min and stored at −80 °C until use. To generate p1 target cells, HEK293T cells were transfected similarly with EBOV trVLP system plasmids but had p4cis-vRNA-Rluc and pCAGGS-T7 replaced with 250 ng pCAGGS-Tim1. Transfected p1 cells were infected with p0 trVLPs, and after 72 h, p1 trVLPs were collected and stored. P2 target cells were prepared similarly and infected with p1 trVLPs. Cells from p0, p1, and p2 were lysed to quantify virus production by Renilla-Glo® Luciferase Assay System (Promega, E2820).

### Authentic EBOV infection

Huh7 cells grown in 24-well plates were transfected with vectors expressing PDIA3, CALR, or CANX, and incubated with EBOV Mayinga strain at 0.01 multiplicity of infection (MOI) for 3 h at 37 °C, which was obtained from National Virus Resource Center, Wuhan Institute of Virology, Chinese Academy of Sciences. After being washed twice with PBS, cells were cultured with fresh medium for indicated times. Viral RNAs were extracted from 140 μL supernatants of each well using QIAamp Viral RNA Kit (QIAGEN 52906) and quantified by Taqman™ Real-Time PCR. The primer pair sequences are 5'-GAGCA TGGTCTTTTCCCTCA-3' [ZEBOV-NP-N-Q (F)] and 5'-TCACGAGACTC CGCATATTG-3' [ZEBOV-NP-N-Q (R)], and the probe sequence is 5'-FAM-TCGCCACAGCCCACGGGAGCACC-3' [ZEBOV-NP-N-Q (P)]. PCR mixtures contained 2 μL viral RNA samples, 5 μL 2x One Step Q Probe Mix, 0.5 μL One Step Q Probe Enzyme Mix (Vazyme, Q222), 0.2 μl 200 nM forward primer, 0.2 μL 200 nM reverse primer, 0.2 μL 200 nM probe, and 1.9 μL RNase-free water. PCR was performed with an CFX96 Real-Time system (Bio-Rad) under the following condition: reverse

transcription at 55 °C for 15 min, pre-denaturation at 95 °C for 30 s, and 40 cycles of amplification (10 s at 95 °C for denaturation, 31 s at 60 °C for annealing and extension). A standard curve was created by titrating a plasmid DNA standard pCAGGS-NP under the exact same PCR conditions, from which viral RNA copy numbers were calculated. qPCR results were analyzed using Bio-Rad CFX Manager 3.1.

### Western blotting

Cells were lysed in ice-cold RIPA lysis buffer (25 mM Tris, pH 7.4, 150 mM NaCl, 0.5% sodium deoxycholate, 0.1% SDS, 1% Nonidet P-40; Sigma-Aldrich, R0278) supplemented with protease inhibitor cocktail (Sigma-Aldrich, P8340). Approximately 0.1 mL buffer was used for a total of $2 \times 10^6$ cells. Cell lysates was centrifuged at $12,000 \times g$ at 4 °C for 10 min. Total protein from virion or cell extracts was boiled in SDS-polyacrylamide gel electrophoresis (SDS-PAGE) loading buffer (Solarbio Life Sciences, P1015) and resolved by SDS-PAGE. Separated proteins were transferred onto PVDF membranes and membranes were blocked with 5% non-fat milk powder in TBST (Tris-buffered saline [20 mM Tris, pH 7.4, 150 mM NaCl] containing 0.1% Tween 20; Solarbio Life Sciences, T8220) for 1 h at room temperature. Membranes were then probed by primary antibody followed by HRP-conjugated secondary antibodies. Chemiluminescence signals were then measured by incubating the membrane with SuperSignal substrate (Thermo Fisher Scientific, 34580).

Mouse monoclonal anti-ERp57 (2881S, 1:1000) and anti-c-Myc (2276, 1:5000) were purchased from Cell Signaling. Rabbit polyclonal anti-Zaire EBOV-GP (40442-T48, 1:5000), anti-EBOV-VP40 (40446-T48, 1:10,000), anti-IAV H5HA (11048-RP02, 1:5000), anti-HDAC6 (100768-T08, 1:1000), and anti-RNF26 (203155-T32, 1:500) were purchased from Sino Biological. Rabbit polyclonal anti-CALR (ADI-SPA-865-F, 1:1000) and anti-CANX (ADI-SPA-600-F, 1:1000) were purchased from ENZO. Mouse monoclonal anti-VSV-G (SAB4200695, 1:1000), anti-FLAG (F3165, 1:5000), anti-HA (H3663, 1:5000), and anti-β-Actin (A5441, 1:5000), and rabbit polyclonal anti-p62 (P0067, 1:3000) and anti-LC3 (L7543, 1:1000) were purchased from Sigma-Aldrich (St. Louis, Missouri). Mouse monoclonal anti-HIV-1 gp120 (521, 1:1000) was obtained from the NIH AIDS Reagent Program (Germantown, Maryland). Rabbit polyclonal anti-LAMP-2A(ab18528, 1:2000) was purchased from Abcam. Rabbit polyclonal anti-GFP (50430-2-AP, 1:5000) and His-HRP (HRP-66005, 1:10,000) were purchased from Proteintech. Horseradish peroxidase (HRP)-conjugated goat anti-mouse IgG (115-035-003, 1:10,000) and anti-rabbit IgG (111-035-003, 1:10,000) were purchased from Jackson ImmunoResearch (West Grove, Pennsylvania).

### Quantification of EBOV-GP on western blots

Gray analyses were performed with ImageJ Launcher 1.4.3.67 (https://imagej.nih.gov/ij/). Brightness and contrast were adjusted equally for all images within a panel. Levels of EBOV-GP were normalized to those of ACTB, and results were obtained from three independent experiments. Results are shown as relative values, with the value from a control (Ctrl) vector set as 1.

### HiBiT blotting

Proteins were similarly separated by SDS-PAGE and transferred to a PVDF membrane. After that, proteins with HiBiT-tag were detected by Nano-Glo® HiBiT Blotting System (Promega, N2410). Briefly, the membrane was incubated with LgBiT/buffer solution (Promega, N2421) for 2 h at room temperature with gentle rocking and then at 4 °C overnight. The membrane was incubated with 20 μL of substrate for 5 min at room temperature and placed between transparent plastic sheets for imaging.

### Immunoprecipitation

After transfection of HEK293T cells cultured in a 10-cm dish, cells were lysed in 0.9 mL RIPA lysis buffer for 30 min on ice. After

removal of nuclei via low-speed centrifugation and collecting 100 μL as input, the remaining 800 μL lysate was incubated with a specific antibody followed by addition of protein G-Sepharose beads (Thermo Fisher Scientific, 22852) and rotated at 4 °C overnight. For target proteins expressing FLAG- and HA-tag, anti-FLAG M2 Magnetic beads (Sigma-Aldrich, M8823) were used for overnight rotation. After being washed 3 times with 1 mL pre-cooled RIPA lysis buffer, proteins were removed from beads after boiling in 40 μL RIPA lysis buffer plus 20 μL sample loading buffer (4×) and analyzed by WB.

### Confocal microscopy

Approximately $1.5 \times 10^5$–$2.0 \times 10^5$ HeLa cells were seeded on glass bottom cell culture dish (NEST Biotechnology Co.Ltd, 801001) and transfected with various vectors using Lipofectamine 3000. After 24 h, cells were fixed with 4% paraformaldehyde, permeabilized with 0.1% Triton X-100 (Solarbio Life Sciences, T8200), and then blocked with 5% bovine serum albumin (BSA; APPLYGEN, P1622) solution. Cells were then incubated with rabbit polyclonal anti-Zaire EBOV-GP (Sino Biological, 40442-T48) at 1:1000 dilution and mouse polyclonal anti-HA (Sigma-Aldrich, H3663) in PBS with 5% BSA for 2 h. After being washed 5 times with PBS, cells were stained with Alexa Fluor 647-conjugated donkey anti-rabbit IgG (Invitrogen, A31573) at 1:700 dilution for 1 h. Cells were washed again with PBS 5 times and then incubated with DAPI for 3 min for nuclear staining. After further washing with PBS, cells were observed under a confocal microscope (LSM880, Zeiss, White Plains, New York). At least 100 random cells per slide were analyzed, and the most representative images from each slide were selected for presentation. Confocal images were analyzed using ZEISS Zen Software version 2.1.

### Graphic preparation

Figures were prepared using Adobe Illustrator 2021. The model in Fig. 10 was created by BioRender (https://biorender.com).

### Statistics and reproducibility

Statistical tests were performed using GraphPad Prism version 9.3.1. Variance was estimated by calculating the standard error of measurements (SEMs) and represented by error bars. All experiments were repeated at least twice, and similar results were obtained. Source data are provided as a Source Data file.

### Reporting summary

Further information on research design is available in the Nature Research Reporting Summary linked to this article.

## Data availability

Source data are provided with this paper.

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

## Acknowledgements

We thank Eric O. Freed and Richard N. Sifers for critical reading and comments on this manuscript. We thank all the colleagues from National Biosafety Laboratory (Wuhan), Chinese Academy of Sciences, for the support and assistant of Ebola virus infection experiments in BSL4 facility. We also thank National Virus Resource Center, Wuhan Institute of Virology, Chinese Academy of Sciences, for providing the Ebola virus stock and Huh7 cells. We thank Ted Dawson, Feng Zhang, and Heinz Feldmann for providing reagents. The model in Fig. 10 was created with BioRender. B.W. is supported by grants from National Natural Science Foundation of China (31873013) and Natural Science Foundation of Heilongjiang Province (QC2017028).

## Author contributions

J.Z. performed all experiments except that using authentic EBOV for infection. B.W. contributed reagents. X.G., C.P., and C.S. conducted the infection experiment with authentic EBOV. S.F.J. created the model. R.C.S. edited the text and provided insightful comments on the paper. Y.H.Z. designed this study and wrote manuscript with input from all coauthors.

## Competing interests

The authors declare no competing interests.
