## [Peer review file · Nature Communications]

REVIEWER COMMENTS

Reviewer #1 (Remarks to the Author):

The manuscript by Zhang et al., describes a potential role of RING finger protein 185 (RNF185) in modifying Ebola GP to downregulate protein levels by reticulophagy. Overall, the manuscript shows in vitro ectopic expressions, knockdowns and pharmacological inhibition studies to support the current conclusions. In fact the data support a potential role of CALR and CANX in GP12 stability. While the studies, as presented may be important, the only biological relevant data provided in the manuscript consists of pseudoviral assays and trVLP assays. Therefore, the main concern associated with this manuscript and the molecular mechanism proposed is the lack of significance and demonstration of impact with Ebola virus. While the reviewer understands that this is a BSL4 virus, given the abundance of reagents generated, including KO cells, testing for viral infection and associated evaluation of GP stability is critical for the premise of this manuscript. Overall, while the data available is supportive of a role for RNF185 in regulating Ebola GP stability via poly ubiquitination, the manuscript is too premature until activity is established under infection conditions. This study should accompany appropriate controls such as viruses shown in Figure 2D.

Reviewer #2 (Remarks to the Author):

The authors present a detailed study that addresses the unusual, and unexpected, molecular players, that orchestrate the elimination of the Ebola viral constituents. I am impressed with the depth and breadth of their experimental design. My significant concern is that the authors don't fully understand the importance of their findings and fail to adequately express this notion. I make this statement because the data contradict what has been considered to represent distinct proteostasis events. My concerns are listed below:

Title and abstract: Most of your readers will not know about reticulophagy which has also been termed ER-phagy, ERLAD, etc. Furthermore, the concepts in the abstract could be significantly broadened. The authors have observed far more than merely determine that the misfolded protein clients (destined for reticulophagy) are ubiquitinated. These issues are discussed below.

Line 21: I would argue that the calnexin-calreticulin cycle does not "promote" degradation. Rather, it is involved in (or contributes to) degradation via reticulophagy. This concept is correctly stated near the end of the manuscript (line 370) when referring to the role of class I mannosidases.

Line 41: There are many kinds of “selected autophagy” systems that do not include reticulophagy. Please provide an explanation for this or used a different term.

Line 73: Again, I would argue that a consequence of the calnexin-calreticulin cycle is to "aid" the elimination of EBOV-GP1,2 by reticulophagy.

Lines 165-177: The hydrolytic activity of class I mannosidases are known to flag glycoproteins for degradation by cytosolic proteasomes. The authors' data imply that they also contribute to the entry of clients into reticulophagy. This conclusion would be considered heresy by many scientists. The findings are VERY IMPORTANT, and controversial because they question current dogma (or they might demonstrate something that has been unknown until now).

Lines 178-199. These sections of the manuscript will be confusing to many readers because the differences and similarities between macroautophagy/autophagy vs. autolysosomes are not defined in the manuscript until the Discussion Session.

Lines 373-374: The traditional thought/conclusion is that the tandem events of class I mannosidases activities and the action of VCP/p97 function to exclusively promote the release of proteins from the endoplasmic reticulum to the cytoplasm for elimination by proteasomes. Therefore, the observation that these components can also play a role in promoting reticulophagy requires an intense level of explanation.

Lines 406-407; 423-426: Considering all that is stated above, it is inadequate, at least in this reviewer's opinion, to simply state “The mechanism for these surprising results remains to be elucidated.” Something must be proposed about a possible explanation for their findings. I would suggest that the authors promote the idea that, as many of us have already concluded, the gradual evolution of the proteostasis system in mammalian cells represents an overall cellular response to viral infections. For this reason, the use of viruses to study proteostasis systems could possibly identify unexpected linkages in the network that were traditionally considered to function separately.

Considering the breadth of the authors' findings, I wonder whether a less focused title would be more appropriate.

Reviewer #1 (Remarks to the Author):

The manuscript by Zhang et al., describes a potential role of RING finger protein 185 (RNF185) in modifying Ebola GP to downregulate protein levels by reticulophagy. Overall, the manuscript shows in vitro ectopic expressions, knockdowns and pharmacological inhibition studies to support the current conclusions. In fact the data support a potential role of CALR and CANX in GP12 stability. While the studies, as presented may be important, the only biological relevant data provided in the manuscript consists of pseudoviral assays and trVLP assays. Therefore, the main concern associated with this manuscript and the molecular mechanism proposed is the lack of significance and demonstration of impact with Ebola virus. While the reviewer understands that this is a BSL4 virus, given the abundance of reagents generated, including KO cells, testing for viral infection and associated evaluation of GP stability is critical for the premise of this manuscript. Overall, while the data available is supportive of a role for RNF185 in regulating Ebola GP stability via poly ubiquitination, the manuscript is too premature until activity is established under infection conditions. This study should accompany appropriate controls such as viruses shown in Figure 2D.

We thank this reviewer for such positive comments on our work. Because we do not have a BSL4 lab that can handle authentic EBOV, it took us a while and great efforts to establish a collaboration with a team of scientists in Wuhan Institute of Virology China who have interests to do these experiments for us. We now can show that PDIA3, CALR, or CANX all inhibit EBOV strain Mayinga replication in a dose-dependent manner in Huh7 cells, which is consistent with our viral assays using HIV-1 pseudoviruses and EBOV trVLPs. These new results are summarized and presented in Fig.2C.

Reviewer #2 (Remarks to the Author):

The authors present a detailed study that addresses the unusual, and unexpected, molecular players, that orchestrate the elimination of the Ebola viral constituents. I am impressed with the depth and breadth of their experimental design. My significant concern is that the authors don't fully understand the importance of their findings and fail to adequately express this notion. I make this statement because the data contradict what has been considered to represent distinct proteostasis events.

We thank this reviewer very much for this important comment!

My concerns are listed below:

Title and abstract: Most of your readers will not know about reticulophagy which has also been termed ER-phagy, ERLAD, etc. Furthermore, the concepts in the abstract could be significantly broadened. The authors have observed far more than merely determine that the misfolded protein clients (destined for reticulophagy) are ubiquitinated. These issues are discussed below.

Line 21: I would argue that the calnexin-calreticulin cycle does not "promote" degradation. Rather, it is involved in (or contributes to) degradation via reticulophagy. This concept is correctly stated near the end of the manuscript (line 370) when referring to the role of class I mannosidases.

The abstract has been completely revised as suggested.

Line 41: There are many kinds of "selected autophagy" systems that do not include reticulophagy. Please provide an explanation for this or used a different term.

It has been changed to ERLAD, ER-phagy, and reticulophagy in the abstract and introduction.

Line 73: Again, I would argue that a consequence of the calnexin-calreticulin cycle is to "aid" the elimination of EBOV-GP1,2 by reticulophagy.

It has been changed to "involved in", as suggested.

Lines 165-177: The hydrolytic activity of class I mannosidases are known to flag glycoproteins for degradation by cytosolic proteasomes. The authors' data imply that they also contribute to the entry of clients into reticulophagy. This

conclusion would be considered heresy by many scientists. The findings are VERY IMPORTANT, and controversial because they question current dogma (or they might demonstrate something that has been unknown until now).
Thanks for this very important comment on these results.

Lines 178-199. These sections of the manuscript will be confusing to many readers because the differences and similarities between macroautophagy/autophagy vs. autolysosomes are not defined in the manuscript until the Discussion Session.

Three types of autophagy are now introduced in this paragraph.

Lines 373-374: The traditional thought/conclusion is that the tandem events of class I mannosidases activities and the action of VCP/p97 function to exclusively promote the release of proteins from the endoplasmic reticulum to the cytoplasm for elimination by proteasomes. Therefore, the observation that these components can also play a role in promoting reticulophagy requires an intense level of explanation.

We have expanded our discussion in this section as suggested.

Lines 406-407; 423-426: Considering all that is stated above, it is inadequate, at least in this reviewer's opinion, to simply state "The mechanism for these surprising results remains to be elucidated." Something must be proposed about a possible explanation for their findings. I would suggest that the authors promote the idea that, as many of us have already concluded, the gradual evolution of the proteostasis system in mammalian cells represents an overall cellular response to viral infections. For this reason, the use of viruses to study proteostasis systems could possibly identify unexpected linkages in the network that were traditionally considered to function separately.
As suggested, we have included these new thoughts in the Discussion at the beginning and the last paragraph.

Considering the breadth of the authors' findings, I wonder whether a less focused title would be more appropriate.

As suggested, we have changed the title to make it much broader to cover the general cell biology.

REVIEWERS' COMMENTS

Reviewer #1 (Remarks to the Author):

This revision is highly constructive and much improved. These studies exploit a novel phenomena and a set of observations that couple Ebola virus biology to proteostasis. All studies are well conducted, appropriate controls are used. Importantly, the authors responded well to the reviews. Overall, this revised version is poised to impact the field with new information and likely lead to additional studies on Ebola viral pathogenesis and the role of proteostasis.

Reviewer #2 (Remarks to the Author):

At the authors' discretion, I recommend some additional modifications to the manuscript that will more greatly enhance the importance of the study's discovery.

Lines 2, 44, 86: Although the term "proteostasis" is often used, "proteostasis network" was the original designation. The use of this more comprehensive term provides support for the authors' observation that unexpected linkages exist in the network of events.

Line 39: More than "one" protein folding cycle operates in the secretory pathway. Therefore, please alter the current use of "one".

Line 57: I suggest that the authors end the sentence with "in this network" rather than insert the idea of a crosslink which is somewhat ambiguous.

Line 357: Begin the sentence with "Responses to viral infections are expected to have played an important role in the evolution of mammalian cells and therefore provide a unique...)

Reviewer #1 (Remarks to the Author):

This revision is highly constructive and much improved. These studies exploit a novel phenomenon and a set of observations that couple Ebola virus biology to proteostasis. All studies are well conducted, appropriate controls are used. Importantly, the authors responded well to the reviews. Overall, this revised version is poised to impact the field with new information and likely lead to additional studies on Ebola viral pathogenesis and the role of proteostasis.

We thank this reviewer for such encouraging comments on our work.

Reviewer #2 (Remarks to the Author):

At the authors' discretion, I recommend some additional modifications to the manuscript that will more greatly enhance the importance of the study's discovery.

Lines 2, 44, 86: Although the term "proteostasis" is often used, "proteostasis network" was the original designation. The use of this more comprehensive term provides support for the authors' observation that unexpected linkages exist in the network of events.

"Proteostasis" has been changed to "proteostasis network" in these three places.

Line 39: More than "one" protein folding cycle operates in the secretory pathway. Therefore, please alter the current use of "one".

"One protein folding cycle" has been changed to "protein folding cycles".

Line 57: I suggest that the authors end the sentence with "in this network" rather than insert the idea of a crosslink which is somewhat ambiguous.

"via direct crosstalk" has been removed.

Line 357: Begin the sentence with "Responses to viral infections are expected to have played an important role in the evolution of mammalian cells and therefore provide a unique...).

It has been included as suggested.